# A quantitative comparison of methods used to measure smaller methane emissions typically observed from superannuated oil and gas infrastructure

Stuart N. Riddick[1], Riley Ancona[1], Mercy Mbua[1], Clay S. Bell[1], Aidan Duggan[1], Timothy L. Vaughn[1], Kristine Bennett[1] and Daniel J. Zimmerle[1]

[1] The Energy Institute, Colorado State University, Fort Collins, CO, 80524, USA

*Correspondence to*: Stuart.Riddick@colostate.edu

**Abstract** Recent interest measuring methane ($CH_4$) emissions from abandoned oil and gas infrastructure has resulted in several methods being continually used to quantify point source emissions less than 200g $CH_4$ hour$^{-1}$. The choice of measurement approach depends on how close observers can get to the source, the instruments available and the meteorological/micrometeorological conditions. As such, static chambers, dynamic chambers, Hi Flow measurements, Gaussian plume (GP) modelling and backward Lagrangian stochastic (bLs) models have all been used, but there is no clear understanding of the accuracy or precision of each method. To address this, we copy the experimental design for each of the measurement methods to make single field measurements of a known source, to simulate single measurement field protocol, and then make repeat measurements to generate an understanding of the accuracy and precision of each method. Here, we present estimates for the average percentage difference between the measured emission and the known emission for three repeat measurements, $A_r$, for emissions of 40 to 200 g $CH_4$ h$^{-1}$. The static chamber data were not presented because of safety concerns during the experiments. Both the dynamic chamber ($A_r$ = -10%, -8%, -10% at emission rates of 40, 100 and 200 g $CH_4$ h$^{-1}$, respectively) and Hi Flow ($A_r$ = -18%, -16%, -18%) repeatedly underestimate the emission, but the dynamic chamber had better accuracy. The standard deviation of emissions from these direct measurement methods remained relatively constant for emissions between 40 and 200 g $CH_4$ h$^{-1}$. For the far field methods, the bLs method generally underestimated emissions ($A_r$ = +6%, -6%, -7%) while the GP method significantly overestimated the emissions ($A_r$ = +86%, +57%, +29%) despite using the same meteorological and concentration data as input. Variability in wind speed, wind direction and atmospheric stability over the 20-minute averaging period are likely to propagate through to large variability in the emission estimate, making these methods less precise than the direct measurement methods. To our knowledge this is the first time that methods for measuring $CH_4$ emissions from point sources between 40 and 200 g $CH_4$ h$^{-1}$ have been quantitively assessed against a known reference source and against each other.

## 1 Introduction

Methane ($CH_4$) gas is a powerful greenhouse gas with a greenhouse warming potential 86 times larger than carbon dioxide over 100 years (IPCC, 2022). Quantification of $CH_4$ emissions from abandoned wells has recently become an area of interest as studies suggest over 200 Gg $CH_4$ yr$^{-1}$ is emitted from 2.2 million abandoned wells in the US alone (US EPA, 2021). Quantifying and then plugging these wells makes them an attractive target for achieving goals set out in the Paris Agreement (Nisbet et al., 2020). Additionally, private companies are beginning initiatives to generate revenue through carbon credits gained by plugging wells and accurate quantification is essential for realizing the capital.

As there are millions of abandoned wells globally, there is a growing need to measure as many wells as quickly as possible to identify the most emissive wells. Typically, an emission from an abandoned well can be considered as an above-ground point

source that is relatively small in emission size, up to 180 g $CH_4$ hour$^{-1}$ (Riddick et al., 2019a; Pekney et al., 2018; Townsend-Small et al., 2016; Boothroyd et al., 2016; El Hachem and Kang, 2022; Saint-Vincent et al., 2020; Townsend-Small and Hoschouer, 2021). Other emission sources, such as emissions from pipeline leakage, are fundamentally different in behavior, where gas travels through the soil and forms an area emission at the surface, these sources require different methods for estimating the emission, e.g.

mass balance or eddy covariance. Area emissions could form if a plugged well leaks from corrosion of the borehole casing, but this will not be discussed in this study.

Several methods are being used to measure emissions from these smaller point sources (less than 200 g $CH_4$ hour$^{-1}$) from abandoned oil and gas infrastructure. The chosen measurement approach depends on how close an observer can get to the source, instrumentation availability and the meteorological/micrometeorological conditions at the measurement site. Measurement

methods can be classed as direct, i.e. touching/enclosing the source, and downwind measurements where access is not possible. Direct methods include static chambers (Livingston and Hutchinson, 1995), dynamic flux chambers (Riddick et al., 2019a, 2020b; Aneja et al., 2006) and Hi Flow sampling (Pekney et al., 2018; Allen et al., 2013; Brantley et al., 2015). While downwind methods include Gaussian-based plume models (Baillie et al., 2019; Caulton et al., 2014; Riddick et al., 2019b, 2020a; Edie et al., 2020; Bell et al., 2017) and Lagrangian dispersion models (Riddick et al., 2019b, 2017; Denmead, 2008; Flesch et al., 1995). Emissions

calculated using the majority of these methods have not been comprehensively compared to controlled emission source rates. Other quantification methods are generally unsuitable for measuring emissions from abandoned wells. While OGI cameras can be used for detecting emissions greater than 20 g $CH_4$ h$^{-1}$ (Ravikumar et al., 2018; Stovern et al., 2020; Zimmerle et al., 2020), using this method for quantification remains in development with few studies published to date investigating the accuracy of emission rate estimates from OGI (Kang et al., 2022). Mass balance approaches are unlikely to detect the small and narrow plume

from the abandoned well. Tracer release is technically demanding, takes a long time to make a single measurement and requires road access for measurement, although it has been used to measure nonproducing wells in Hungary (Delre et al., 2022). Remote sensing has typical detection limits of 10+ kg $CH_4$ h$^{-1}$ for aircraft (Duren et al., 2019), 100+ kg $CH_4$ h$^{-1}$ for satellites (Cooper et al., 2022) and unsuitable for these types of emission source. As such, these other quantification methods will not be investigated in this study.

In general, as access becomes more restricted, emission rates larger, or safety concerns increase (such as the co-emission of harmful gases), the method used to estimate the $CH_4$ emission rate of a source must be carefully considered. From experience and the response of a 4-gas monitor, working close enough to measure emissions greater than 200 g $CH_4$ h$^{-1}$ for many of these methods (especially the chambers and Hi Flow) can be unsafe, therefore this study is limited to quantifying $CH_4$ emissions between the lowest flow METEC can produce (40 g $CH_4$ h$^{-1}$) and the highest flow we feel comfortable measuring with these methods (200 g

$CH_4$ h$^{-1}$). Putting these emission ranges into real-word context, the maximum emission from unplugged and abandoned wells was measured at 177 g $CH_4$ h$^{-1}$ in West Virginia (Riddick et al., 2019a), 175 g $CH_4$ h$^{-1}$ in Pennsylvania (Pekney et al., 2018), 146 g $CH_4$ h$^{-1}$ across the US (Townsend-Small et al., 2016) and 35 g $CH_4$ h$^{-1}$ in the UK (Boothroyd et al., 2016). As most of the methods presented here require access to the source, we considered 200 g $CH_4$ h$^{-1}$ to be a sensible limit to the emission rate and is larger than the emissions observed by many previous studies. Therefore, the scope of this study is limited to estimating $CH_4$ emissions

from a single point source that we would realistically be able to approach and measure, i.e. between 40 and 200 g $CH_4$ h$^{-1}$. The study compares each method's accuracy against known emission rates. Explicitly, our objectives are: 1) Reproduce the experimental design for each of the measurement methods; 2) Conduct repeat measurements, as a researcher would do in the field, by taking measurements to generate an emission estimate from a point source and compare this to known emission rate; and 3) Make recommendations on the suitability of each method for measuring emissions from relatively small point sources. We add

the caveat that we will only present data from measurement methodologies conducted safely wearing personal protective equipment

(PPE) as regulated at the Colorado State University Methane Emissions Technology Evaluation Center (METEC) facility in Fort Collins, CO, USA (steel toe boot, flame resistant (FR) overalls, hard hat, safety glasses and 4-gas monitor). To our knowledge this is the first time that methods for measuring $CH_4$ emissions from point sources between 40 and 200 g $CH_4$ h$^{-1}$ have been quantitively assessed against a known reference source and against each other.

**2 Methods**

Each of the methods, static chambers, dynamic chambers, Hi Flow, bLs and GP, are tested at METEC in Fort Collins, CO, USA. METEC can reproduce the range of $CH_4$ emissions typically seen from individual point sources at oil and gas operations, i.e. between 20 g $CH_4$ hr$^{-1}$ and 40 kg $CH_4$ hr$^{-1}$, from realistic locations on O&G equipment. At the METEC site, compressed natural gas, with methane compositions ranging from 85 to 95%vol, is supplied from two 145 L cylinders and flow rates controlled using

a pressure regulator and precision orifices. At METEC the methane content of the natural gas in each release is measured by gas chromatography and accounted for in the known emission rate. For the purposes of this study, where we are comparing the ability of each method to estimate the emission from a point source, we will constrain the known emission rates to those that can be measured safely, i.e. between 40 and 200 g $CH_4$ hr$^{-1}$. To accomplish this, $CH_4$ emission rates will be set from a point source (diameter 6 mm) at 20 cm above the ground at 40, 100 and 200 g $CH_4$ hr$^{-1}$.

Two instruments are used to report $CH_4$ mixing ratios: the Picarro (ww.picarro.com) GasScouter G4301 mobile gas concentration analyser and the Agilent (www.aglient.com) 7890B Flame-Ionization Detector Gas Chromatograph (GC-FID). The Picarro GasScouter reports $CO_2$, $H_2O$ and $CH_4$ mixing ratios every 3 s, with a precision (300s, 1$\sigma$) for $CH_4$ of 300 ppb over an operating range of 0 to 800 ppm. The Agilent 7890B gas chromatograph-flame ionization detector (GC-FID), as used here, has a detection limit of 1.5 ppb and linear dynamic range from 1 ppm to 100% $CH_4$. The instrument was calibrated every 10 samples using a

5,000 ppm $CH_4$ gas standard (accuracy of standard ± 5%). The GC-FID was checked for linearity before and after each set of measurements using zero-air, 5,000 ppm, 2.5% and 100% $CH_4$.

**2.1 Static Chamber**

For the static chamber method, a container of a known volume ($V$, m$^3$) is placed over the emission source and the change in concentration ($C$, g m$^{-3}$) inside the container over time ($t$, s) can be used to calculate the emission ($Q$, g s$^{-1}$; Equation 1). The static

chamber method requires no power and is very portable. The major shortcoming of this method is that large emission sources can result in the concentration inside the chamber exceeding the $CH_4$ lower explosive limit (LEL).

$$Q = \frac{dC}{dt}.V \qquad \text{(Equation 1)}$$

Following method descriptions presented in Kang et al. (2014), the static chamber is made by enclosing air within a fixed volume over the emission source (Figure 1A). The chamber was constructed of two parts, a smaller lower part that was secured 4 cm into

the soil and a larger upper part that was fixed to the lower part at the start of the experiment. A fan was secured inside the chamber and used to circulate the air to ensure the air inside the chamber was fully mixed (Kang et al., 2014, 2016). As the experiment was conducted at METEC, 120 V mains power was used, however, in a remote locations power can be supplied by anything capable of delivering a stable 12 V output (e.g. battery). When the chamber is sealed with the ground, following Kang et al. (2014; 2016), an air sample is drawn using a gas syringe. During the experiment air samples are taken at regular intervals, with the time interval

pre-calculated depending on the emission rate to ensure the increase in concentration was linear. The emission is then calculated from the linear increase in concentration over time.

Two sizes of static chambers were used in this experiment ($0.12$ m$^3$ and $0.5$ m$^3$; Figure 1), the chambers were made from rigid plastic cylindrical chambers, with heights approximately 1.5 times the chamber's diameter. The chamber sizes were based on a measurable concentration change over time for given release rates, however, it is unlikely that the larger size is practical for field deployment. During any wind the chamber acted as a sail and the larger ($0.5$ m$^3$) chamber lifted from the ground, therefore, smaller chambers are better in the windy conditions but quickly fill with gas making quantification difficult as the change in CH$_4$ concentration inside the chamber quickly becomes non-linear. In each case, the chamber was placed over a point source 20 cm above the ground emitting gas at approximately 40, 100 and 200 g CH$_4$ hr$^{-1}$. During the experiment, four samples of 25 ml of air were drawn from the chamber using a 50 ml gas syringe at equal time intervals (Pihlatie et al., 2013; Collier et al., 2014). The air samples were injected into glass vials containing 30 ml of nitrogen and then stored in a fridge before the CH$_4$ concentrations were measured using the GC-FID. All samples were measured within two hours of collection. All experiments were repeated three times.

The minimum time between air sampling was set at one minute to ensure that the correct vial could be found, and the sample outlet purged of gas. When sampling times were less, the experiment became too rushed and errors occurred. Additionally, as a health and safety precaution, a handheld CH$_4$ sensor, HXG-2D (Sensit Technologies, USA, www.gasleaksensors.com; detection limit 10 ppm and range 0 to 40,000 ppm), was placed in the chamber and if the CH$_4$ concentration exceeded the lower explosive limit before three samples were taken the test was abandoned.

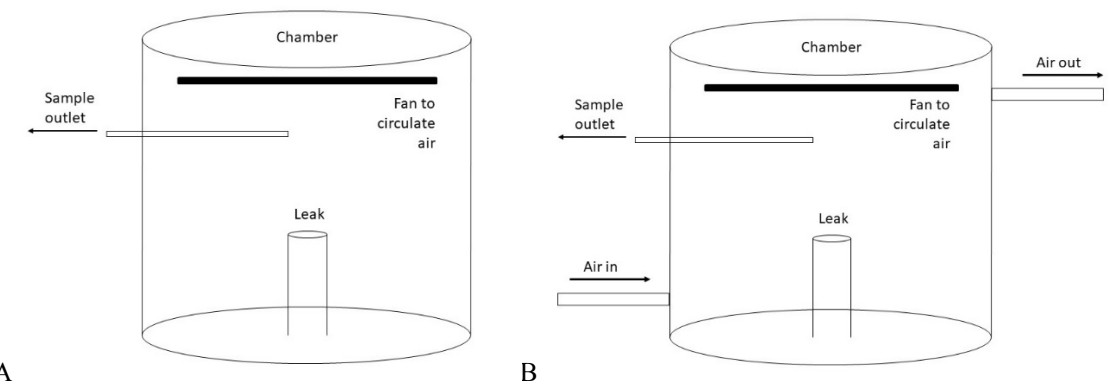

A                                                              B

**Figure 1 Schematics of the A. Static chamber and B. Dynamic flux chamber.**

**2.2 Dynamic Chamber**

To address LEL issues inside the chamber, a dynamic flux method has also been used to measure CH$_4$ leakage from abandoned and active oil and gas wells (Riddick et al., 2019a). Like the static chamber, the dynamic chamber comprises of a container ($0.12$ m$^3$) enclosing the source and a propeller is used to circulate the air. Additionally, a flow of air is passed through the chamber, which reduces the likelihood of exceeding LEL inside the chamber. Unlike the static chamber, the CH$_4$ concentration becomes stable after a period of time depending on the source emission rate. When the chamber reached steady state, three air samples were taken from inside the chamber. A background air sample was taken outside the chamber as the chamber approached steady state. The methane concentration in all air samples was measured using a gas chromatography. The CH$_4$ flux ($Q$, g s$^{-1}$) is calculated (Equation 2) from the CH$_4$ concentration at steady state ($C_{eq}$, g m$^{-3}$), the background CH$_4$ concentration ($C_b$, g m$^{-3}$) in the air used to flush the chamber, the height of the chamber ($h$, m), the flow of air through the chamber ($q$, m$^3$ s$^{-1}$), the footprint of the chamber ($a$, m$^2$) and the volume of the chamber ($V$, m$^3$) (Aneja et al., 2006; Riddick et al., 2019a). As well as improving the safety, the

dynamic chamber reduces the theoretical uncertainty in emission rate to ± 7% (Riddick et al., 2019a), however, the added power requirement of a pump means the dynamic chamber is less portable than the static chamber. Methane emissions from abandoned wells have been quantified using this method between 4 µg $CH_4$ $hr^{-1}$ and 100 g $CH_4$ $hr^{-1}$ (Riddick et al., 2019a).

$$Q = \frac{(C_{eq} - C_b) \cdot h \cdot q \cdot a}{V} \qquad \text{(Equation 2)}$$

A single chamber 0.12 $m^3$ was used for testing the dynamic chamber method. The plastic chamber, open at one end, was placed over known leaks of approximately 40, 100 and 200 g $CH_4$ $hr^{-1}$ and air was passed through the chamber at a constant rate of 67 l $min^{-1}$, following the method of Riddick et al. (2019). As the experiment was conducted at METEC, 120 V mains power was used, however, in a remote location power can be supplied by anything capable of delivering a stable 12 V output. The chamber was left until the $CH_4$ concentration inside had become constant, as measured by a Sensit HXG-2D sensor (Sensit Technologies,

Valparaiso, IN, USA). When steady state was reached, three sample of 25 ml of air were drawn from the chamber using a 50 ml gas syringe injected into glass vials containing 30 ml of nitrogen. As with the samples from the static chamber, the vials were measured within two hours of collection. All experiments were repeated three times. Following the methods of Aneja et al. (2006) and Riddick et al. (2019; 2020), the emission rate is calculated from the steady state gas concentration using Equation 2.

## 2.3 Hi Flow

Another way of addressing the issue of enclosing methane at concentrations approaching LEL is to use a Hi Flow sampler. A Hi Flow sampler draws high volumes of air into a measurement chamber at a fixed rate ($F$, $m^3$ $s^{-1}$) and the background $CH_4$ concentration ($X_b$, g $m^{-3}$) and the concentration of $CH_4$ in the air is measured ($X_s$, g $m^{-3}$) are used to calculate the emission rate ($Q$, g $s^{-1}$) (Equation 3). The Bacharach Hi Flow Sampler (Bacharach, Pittsburgh, USA, www.mybacharach.com) is the only currently-available Hi Flow sampler and was used in this study, it draws air at between 226 and 297 l $min^{-1}$ and can measure $CH_4$ emissions

between 50 g $CH_4$ $h^{-1}$ to 9 kg $CH_4$ $h^{-1}$ to an accuracy of ± 10% (Connolly et al., 2019). A recent study commissioned by the California Air Resources Board developed open-source architecture for a new Hi Flow unit which is capable of replacing the current Bacharach Hi Flow Sampler (Vaughn et al., 2022).

$$Q = F. (X_s - X_b) \qquad \text{(Equation 3)}$$

As the Hi Flow sampler method is relatively simple, no data is required other than the direct measurements made by the instrument.

Following the methods of Pekney et al. (2018), the bag containing the hose end of the Bacharach Hi Flow sampler was placed over the point source and the instrument was turned on. This was repeated three times and the average emission calculated. The Bacharach Hi Flow sampler used in this study was calibrated monthly as recommended by the manufacturer.

## 2.4 Gaussian Plume

In some circumstances, access and safety restrictions mean that direct measurements are impossible, and an observer must use a

far-field method to measure the emissions remotely. The most widely used of these far-field approaches is the Gaussian plume (GP) model. First used in the 1940s, a GP model describes the concentration of a gas as a function of distance downwind from a point source (Seinfeld and Pandis, 2016). When a gas is emitted from the source, it is entrained in the prevailing ambient air flow and disperses laterally and vertically with time, forming a dispersed concentration cone. The concentration enhancement of the gas ($X$, µg $m^{-3}$), at any point $x$ meters downwind of the source, $y$ meters laterally from the center line of the plume and $z$ meters

above ground level can be calculated (Equation 4) using the emission rate ($Q$, g $s^{-1}$), the height of the source ($h_s$, m) and the Pasquill-Gifford stability class (PGSC) as a measure of air stability. The standard deviation of the lateral ($\sigma_y$, m) and vertical ($\sigma_z$, m) mixing ratio distributions are calculated from the PGSC of the air (Pasquill, 1962; Busse and Zimmerman, 1973; US EPA, 1995). The GP

model assumes that the vertical eddy diffusivity and wind speed are constant and there is total reflection of $CH_4$ at the surface, where gas reflected from the surface of the Earth is accounted for in the downwind plume. The enhancement is defined as the difference between the downwind concentration and the background concentration measured upwind. The GP is the simplest of the far-field methods considered here and assumes that the emissions are well-defined plumes injected above the near-surface turbulent layer from point sources and not affected by aerodynamic obstructions that cause mechanical turbulence at the surface. However, in most situations there are aerodynamic obstacles and plumes are rarely perfectly Gaussian in shape. Another shortcoming of the GP model is the parameterization of the PGSC, which are discrete values and incorrectly assigning them can lead to significant uncertainty. Generally, speaking the GP is rarely used for emissions less than 100 g $CH_4$ $h^{-1}$. However, an example of using a GP model is its use in estimating $CH_4$ emissions from oil production platforms in the North Sea, where emissions ranged from 10 to 80 kg $CH_4$ $hr^{-1}$ with an uncertainty of $\pm 45\%$ (Riddick et al., 2019b).

$$X(x, y, z) = \frac{Q}{2\pi u \sigma_y \sigma_z} e^{-\frac{y^2}{(2\sigma_y)^2}} \left( e^{-\frac{(z-h_S)^2}{(2\sigma_z)^2}} + e^{-\frac{(z+h_S)^2}{(2\sigma_z)^2}} \right) \qquad \text{(Equation 4)}$$

The GP model uses downwind measurement coupled with meteorology to estimate the emission rate of a source using equation 4. Explicitly, the data used are wind speed ($u$, m $s^{-1}$), wind direction ($WD$, °), temperature ($T$, °C), $CH_4$ concentration downwind of the source ($X$, µg $m^{-3}$), location and height of the $CH_4$ detector, background $CH_4$ concentration ($X_b$, µg $m^{-3}$) and the PGSC. The PGSC can either be calculated using the wind speed and a measure of the solar irradiance (Supplementary Material Section 1 Table S1) or using a sonic anemometer. Due to power requirements, sonic anemometers are unlikely to be used in the field and, as such, a more basic approach is adopted and the PGSC calculated from the wind speed ($u$, m $s^{-1}$) measured at 1.2 m and irradiance measured at the emission point ($G$, kW $m^{-2}$). Pasquill and Smith (1983) originally defined strong irradiance as sunny midday in midsummer in England and slight insolation to similar conditions in midwinter. Here we classify strong irradiance as > 1 kW $m^{-2}$, moderate irradiance 0.5 kW $m^{-2}$ to 1 kW $m^{-2}$ and light irradiance as > 0.5 kW $m^{-2}$ (Riddick et al., 2022).

Methane emissions are calculated using $CH_4$ concentrations measured 1.5 m above ground level, 5 m downwind and background $CH_4$ concentrations 5 m upwind of the source by the Picarro GasScouter. Here, it assumed that the experiments are conducted as close as possible to the source (between 1 and 10 m) without direct access to the emission point. Wind speed and wind direction were measured every 10 s using a Kestrel 5500 weather meter (www.kestrelmeters.com) on a mast 2 m above the ground. To reduce any impact of mechanical turbulence while maintaining real changes to $CH_4$ emission caused by changing environmental or atmospheric factors, both $CH_4$ concentrations and meteorological data are averaged over 15 min (Laubach et al., 2008; Flesch et al., 2009). The PGSC was calculated from the meteorological data using the method of Seinfeld and Pandis (2006). The lookup table, Table S1, is presented in Supplementary Material Section 1. Complex topography, such as building and trees, are not parameterized or accounted for by the GP model.

**2.5 bLS dispersion model point measurements**

As an alternative to the GP model, Lagrangian dispersion models can be used to calculate the emission of a source. In a backward Lagrangian stochastic (bLS) model, the measurement position, gas concentration, meteorology and micrometeorology are known inputs and the model works iteratively backwards to simulate the motion of the air parcel, this is then used to infer the rate of emission from the source (Flesch et al., 1995). For given meteorological conditions, the model calculates the ratio of downwind concentration to emission, $(C/Q)_{sim}$, depending on the size and location of the source. The emission rate ($Q$, g $m^{-2}$ $s^{-1}$) is then inferred from the measured gas concentration at 1.2 m above ground level ($X_m$, g $m^{-3}$) and the background gas concentration ($X_b$, g $m^{-3}$) (Equation 5). The bLS models can be used to calculate the emissions from point or area sources in a range of micrometeorological conditions. However, a major shortcoming of the model is its inability to adequately model emissions from

sources with complex topography or near large objects, such as buildings. This can be mitigated by measuring far away from the source over a relatively flat fetch, but an accurate measurement of the micrometeorology is required. As an example, $CH_4$ emissions from individual point sources on oil and gas infrastructure have been estimated using a bLS model between 4 µg $CH_4$ $hr^{-1}$ and 3 kg $CH_4$ $hr^{-1}$ with an uncertainty of ± 38% (Riddick et al., 2019a)

$$Q = \frac{X_m - X_b}{\left(\frac{C}{Q}\right)_{sim}}$$
(Equation 5)

WindTrax (www.thunderbeachscientific.com), a commercial software program, uses a bLS dispersion model to calculate the rate of gas emission from a point, area or line source. In this application, the inversion function of the WindTrax inverse dispersion model version 2.0 was used (Flesch et al., 1995). Data used as input are wind speed ($u$, m $s^{-1}$), wind direction ($WD$, °), temperature ($T$, °C), downwind $CH_4$ concentration ($X$, µg $m^{-3}$), location and height of the $CH_4$ detector, background $CH_4$ concentration ($X_b$, µg $m^{-3}$), the roughness length ($z_0$, m) and the Pasquill-Gifford stability class. The ideal terrain for WindTrax modelling is an obstruction-free surface (Sommer et al., 2005; Laubach et al., 2008) with the maximum distance between the source and the detector of 1 km (Flesch et al., 2005, 2009). The roughness length was set at 2.3 cm to represent the short grass of the fetch. Again, it assumed that the experiments are conducted as close as possible to the source without direct access to the emission point. Data for downwind average $CH_4$ concentration, background $CH_4$ concentration, meteorological and micrometeorological data used in WindTrax will be the same as described in Section 2.4.

## 2.6 Measures of accuracy and precision

In each individual experiment the difference between the known emission rate and the calculated emission rate will be presented as a percentage (Equation 6), where $A$ is the accuracy, $Q_c$ is the calculated emission and $Q_k$ is the known emission. The average accuracy of the three experiments ($A_r$, %) will be presented as a measure of the accuracy and the standard deviation ($A_{S.D.}$) of the individual uncertainties will be used as a comparative measure of the precision.

$$A = \frac{(Q_c - Q_k)}{Q_k} x\ 100$$
(Equation 6)

## 3 Results

### 3.1 Method narrative – Qualitative observations of methods

The static chamber is fixed around an emission source and extracts air samples at known time intervals. These vials can be stored for up to a month before analysis on a gas chromatograph. As such, the samples can be analyzed by a third party and the researcher only requires access to the flux chamber, LEL sensor, and the required gas sampling equipment. We found the main shortcomings of the static chamber method are: 1. It was difficult to take samples fast enough during the linear change in concentration; and 2. The method is inherently dangerous as we were unable to remove the chamber without the four-gas monitor, worn on the observer's collar, detecting $CH_4$ concentrations that exceeded the lower explosive limit, i.e. triggered the monitor's alarm.

To address the first shortcoming, a trace gas analyzer could be used to measure the concentrations inside the chamber. As trace gas analyzers use a pump to draw air into the measurement cavity, the analyzer could be arranged in one of two ways. Both introduce additional uncertainty into the quantification. If the gas is removed from the chamber (i.e. the air from the analyzer exhaust is actively pushed outside the chamber), the static chamber becomes a dynamic chamber and the analyzer flow rate must be accounted for in the quantification. If the measured gas is reintroduced to the chamber (i.e. the analyzer outlet is vented back to the chamber), a gas of lower concentration is being continually added to the "closed" system and it is therefore unclear how

much uncertainty is caused by this cycling. Furthermore, the linear response of a portable trace gas analyzer, e.g. the ABB GLA131-GGA Greenhouse Gas Analyzer (https://new.abb.com/), is 100 ppm. Using the lowest emission rate in the study, 40 g $CH_4$ h$^{-1}$, and the largest chamber, 0.5 m$^3$, the concentration inside the chamber will exceed the linear range within 7 seconds. Another alternative could be using a lower precision sensor with a larger detection range, such as the SGX INIR- ME100 (https://sgx.cdistore.com/) that can measure from 200 ppm to 100% methane by volume (bv), but safety issues remain.

We were aware throughout the experiment that the chamber will become explosive and pre-calculated the time between sample measurement based on the emission rate. During the 200 g $CH_4$ h$^{-1}$ experiment, the lower explosive limit of $CH_4$ was reached after three minutes of the chamber being sealed. As such, we have not presented the measurement data collected during the static chamber experiments and strongly encourage the use of an alternative method. The static chamber could be automated to release gas when $CH_4$ concentration inside the chamber approaches LEL to prevent chamber becoming explosive. The major shortcoming of this strategy is that the automation of a chamber takes away the operator's control of when gas is released, which could happen at an inconvenient time during measurement. If an automated system is used for collecting gas of unknown composition self-contained breathing apparatus should be worn.

**Table 1 Condensed description of logistical needs and results of each experiment. *Access* describes if physical access to the emission source is required (Y denotes having permission to touch/enclose the emission point and N denotes experiments are conducted as close as possible to the source without direct access), *Inst* describes if a dedicated instrument is required, and *Cost* is the approximate cost of the lowest price instrument capable of the measurements. *Met* describes if meteorological data is required. *$T_{meas}$* and *$T_{analysis}$* are the times it takes to conduct and analyse one measurement, respectively. *A* is the accuracy of one measurement of a 200 g $CH_4$ h$^{-1}$ source (as defined above in Section 2.6), *$A_r$* is the average accuracy when repeating the measurement of a 200 g $CH_4$ h$^{-1}$ source three times, *$A_{S.D.}$* is the standard deviation of the accuracy of the three repeated experiments and *U* is the theoretical uncertainty as presented in previous studies.**

| Method | *Access* | *Inst* | *Cost* (k$) | *Met* | *$T_{meas}$* (mins) | *$T_{analysis}$* (mins) | *A* (%) | *$A_r$* (%) | *$A_{S.D.}$* | *U (%)* |
|---|---|---|---|---|---|---|---|---|---|---|
| Static chamber | Y | N | ◊ | N | - | - | - | - | - | - |
| Dynamic chamber | Y | N | ◊ | N | 15 | 5 | -11 | -10 | 5.9 | ± 7[#] |
| Hi Flow | Y | Y | 35 | N | 5 | - | -16 | -18 | 8.2 | ± 10[†] |
| Gaussian Plume | N | Y | 32 | Y | 15 | 60 | 33 | 29 | 12.5 | ± 18[‡] |
| bLs model | N | Y | 32 | Y | 15 | 90 | -11 | -7 | 14.1 | ± 12[§] |

[#] Riddick et al. (2019), [†] Pekney et al. (2015), [‡] Riddick et al. (2020), [§] Riddick et al. (2016)

- the static chamber data is not presented as the method was found to be inherently dangerous.

◊ Cost of sample analysis by GC will vary by laboratory.

The dynamic chamber is logistically one step more advanced than the static chamber and requires a pump to draw air through the chamber at a known rate, and, ideally, a flow meter to measure the air flow. This reduces the potential for $CH_4$ concentration inside the chamber becoming explosive. This means the main advantages of the static chamber are preserved, i.e. cost and ease of analysis, but mitigates the health and safety concerns. Again, the major shortcoming of the dynamic chamber method is that it requires direct access to the emission source and a 12 V power source for the pump. Another factor that could affect accuracy of measurement is the air being pumped into the chamber, care should be taken to ensure the inlet is apart from other $CH_4$ sources and far away from the chamber outlet.

The Hi Flow is an off-the-shelf method/instrument, and as an integrated solution, is easier than the dynamic chamber. Once calibrated, the Hi Flow bag is loosely cinched around the emission source and turned on. The instrument displays the methane emission, in l min$^{-1}$, within a minute at a precision of one significant figure. The data are stored in the instrument and can be downloaded later. The advantages of the Hi Flow are the ease of use and amount of time needed to measure a source, typically five minutes per emission source. The main shortcomings are that the researcher needs to have a Hi Flow instrument (costs $35,000), direct access to the source, calibration gas, and a means of charging batteries and/or powering the instrument.

Measurement data required for the GP and bLs methods were the same. After $CH_4$ is emitted from a source it quickly disperses and to measure the concentration downwind access to a sub-ppm $CH_4$ analyzer is required. In 2020, the least-expensive, suitable instrument on the market costs around $32,000. In addition to near-ambient $CH_4$ concentration measurements, meteorological data are required to populate the models. Despite the cost and time required to make the measurements, the practical advantages of these methods are that access is not required and emissions can be calculated from remote sources. However, ensuring that the measurement location is in the plume for long enough to detect an enhancement large enough for the instrument to measure accurately can be challenging. In light winds the plume can move laterally and the sensor becomes offset.

**3.2 Accuracy and precision of repeat measurements**

Static chamber results are not presented as we were unable to remove the chamber without exposing the observer to an explosive environment. Our results show that the most accurate method for generating emissions after repeat measurements from a 200 g $CH_4$ h$^{-1}$ source was the bLs method (-7%), then the dynamic chamber (-10%) and then the Hi Flow (-18%) (Table 1). The least accurate method after repeat measurements was the GP model (29%). Repeating the experiments improved the accuracy of the emission estimate by 4% for the GP model. Data are all presented in Supplementary Material Section 3. For the 40 g $CH_4$ h$^{-1}$ source, repeating the experiments generally improved the accuracy of the emission estimate except for the GP model which became 20% less accurate (Figure 2A). Like the accuracy, the precision of the methods became better, i.e. the standard deviation (S.D.) of the individual uncertainties became smaller, as the emission rate of the source increased (Figure 2B). Methods that made measurements while being attached to the source – chamber and Hi Flow methods – were more precise than those that measured remotely – bLs and GP methods.

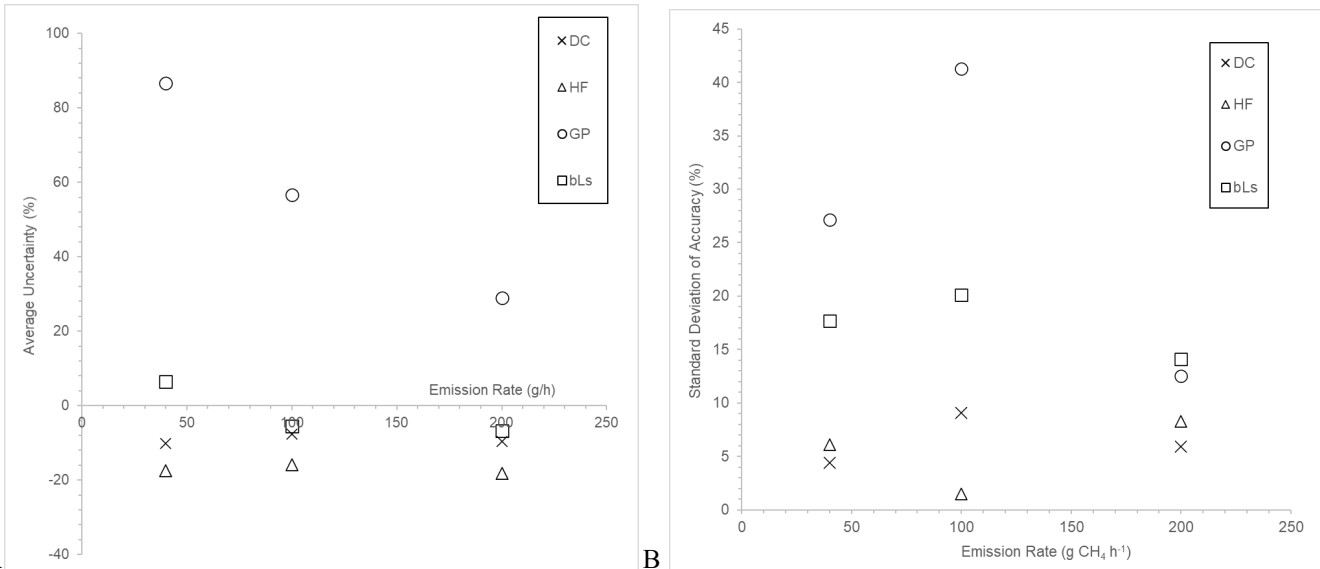

**Figure 2 A) Average accuracy (% difference from known emission rate) of emission estimates from three repeat measurements using each of the measurement methodologies at different known emission rates (~ 40, 100 and 200 g CH$_4$ h$^{-1}$). B) The standard deviation of the uncertainties of repeated measurements against the emission rate of the experiment. Abbreviations as follows: DC – Dynamic chamber, HF – Hi Flow, GP – Gaussian Plume, bLs – Backwards Lagrangian stochastic method.**

## 4 Discussion

This study investigates the utility, accuracy and precision of five methods that have recently been used to estimate smaller, < 200 g CH$_4$ h$^{-1}$, CH$_4$ emissions from oil and gas infrastructure and include, the static chamber, dynamic chamber, the Bacharach Hi Flow sensor, Gaussian plume modelling and backward Lagrangian stochastic models. When the method has been shown to be no danger to the observer, we generate CH$_4$ emission estimates from a known CH$_4$ source emitting approximately 40, 100 and 200 g CH$_4$ h$^{-1}$. Experiments simulating published methods are carried out once to generate a single visit estimate and are then repeated twice more to better understand how repeat experiments can improve the accuracy and precision of the emission estimate.

The static chamber method was found to be inherently dangerous, as the observer was unable to remove the chamber without being exposed to an explosive environment. As a result, the data from the static chamber experiments have not been presented in this study. Furthermore, the experiment conducted at METEC used processed natural gas where heavier/aromatic hydrocarbons and toxic gases have been removed. Gas emitted from abandoned oil and gas wells is unrefined and we advise that the static chamber method should not be used to quantify emissions of an unknown composition of natural gas as this could expose the observer to high concentrations of toxic gas. Therefore, we recommend that one of the other methods presented here should be used to quantify emissions from abandoned oil and gas wells.

Both the dynamic chamber ($A_r$ = -10%, -8%, -10% at emission rates of 40, 100 and 200 g CH$_4$ h$^{-1}$, respectively) and Hi Flow ($A_r$ = -18%, -16%, -18%) repeatedly underestimate the emission, but the dynamic chamber is more accurate. For the far field methods, the bLs method underestimated emissions ($A_r$ = +6%, -6%, -7%) while the GP method significantly overestimated the emissions ($A_r$ = +86%, +57%, +29%) despite using the same meteorological and concentration data as input. These findings are consistent with another study (Bonifacio et al., 2013), however, this is the first study that has compared both to a known emission rate. In all cases the accuracy in the emission estimate increased with emission rate apart from the Hi Flow. The Bacharach Hi Flow system is designed to measure emission from 50 g CH$_4$ h$^{-1}$ to 9 kg CH$_4$ h$^{-1}$ to an accuracy of ± 10%. All flow rates presented here are at

the lowest range that the Hi Flow can measure, and it is likely that the uncertainty in the systems sensors that measures between 40 and 400 g $CH_4$ $h^{-1}$ is of negligible difference.

The method that improves the most as the emission rate increases is the GP method, where accuracy increases from +87% to +29% as the emission rate increased from 40 to 200 g $CH_4$ $h^{-1}$. This improvement in emission is likely caused by the increased size of the plume and the ability of GP model to parameterize the concentration at distances from the centerline of the plume. Although not explicitly stated, the parameterization of the lateral dispersion in the GP model is the same at 100 m as at 5 m which is unlikely. Other controlled release experiments using the GP approach show similar uncertainties, one experiment reported average emissions calculated using a GP model less than 20% (release rates were not published), with the uncertainty mainly driven by atmospheric variability (Caulton et al., 2019). Another showed uncertainties of ±50% for triplicate measurements of emissions between 90 and 970 g $CH_4$ $h^{-1}$ (Caulton et al., 2018).

Data do not exist on controlled release experiments using a dynamic chamber. One study suggested a theoretical emissions uncertainty in the dynamic chamber approach of ±7% (Riddick et al., 2019a), with the largest source of uncertainty caused by the measurement of the flow rate of air through the chamber. Other sources of uncertainty for the dynamic chamber methods are relatively negligible as the methane quantification of the background gas and the gas at steady state (assuming complete mixing of the gas in the chamber) using the GC is highly accurate over a large concentration range and the volume of the chamber fixed by a plastic structure.

A controlled release has been conducted for the bLs model, but only for an emission from an area source (Ro et al., 2011) at the surface and not analogous to the emissions of this study. Ro et al. (2011) estimated the bLs uncertainty at ± 25% for a gas emitted at an unspecified rate from a 27 $m^2$ emission area. As with the GP approach, the bLs model's main source uncertainty is the parameterization of the atmospheric stability (Riddick et al., 2012; Flesch et al., 1995; Ro et al., 2011). The main advantage of the bLs model over the GP at these short distances is it calculates the lateral dispersion of gas for individual particles, while the GP uses an averaged dispersion parameter.

The emission estimates quantified using direct methods, dynamic chamber and Hi Flow sampler, have a lower S.D. than the far-field methods (Figure 2B). The S.D. of direct measurement methods remain relatively constant for emissions between 40 and 200 g $CH_4$ $h^{-1}$ and reflects the relative simplicity of the methods. Assuming all other parameters are measured correctly, for direct methods the variability in emission estimate is a function of how well the $CH_4$ is mixed into the air in the chamber during the measurement.

Variability in the far field emission estimates is much larger and reflects the relative complexity of inferring emissions. Variability in wind speed, wind direction and atmospheric stability over the 20-minute averaging period are likely to propagate through to large variability in the emission estimate. It may be reasonable to suggest that the variability in bLs calculated emission is less than for the GP method because of the added parametrization available (roughness length and gas species). In summary, the penalty of downwind measurement is a higher uncertainty in individual measurements, but this appear to be corrected for by the bLs model through repeat measurements where uncertainty is corrected for by the stochastic nature of particle movement modelling.

Regardless of accuracy and precision, this study shows that all methods can be used to estimate emissions from a source between 40 and 200 g $CH_4$ $h^{-1}$ to an accuracy of at least 40%. It is reasonable to assume that this level of uncertainty is acceptable in some studies where the research is only aiming to determine relative sizes of emission, e.g. Riddick et al. (2019), while other studies require time-resolved emission estimates to compare against modelled output, e.g. Riddick et al. 2017.

It is, however, concerning that many of the methods show a bias in measurement results and in particular the GP model (Figure 3). In most studies, it is assumed that in taking multiple measurements the average uncertainty will be reduced to an aggregate, unbiased emission estimate. Taking the GP emission estimates as an example, the individual calculated emissions are all overestimates of the true emission, therefore, suggesting a fundamental shortcoming in the method (Figure 3). These measurements were taken four days apart in similar environmental conditions (all PGSC C) with wind direction being the only difference between measurements, which can be seen from the correlation between the uncertainty and horizontal distance from plume center (Figure 3B). As mentioned above, it is likely that this is due to the lateral dispersion in the GP approach being parametrized incorrectly, i.e. using values that were defined for distances of 100 m. This suggests that using the GP approach with a single measurement in the plume for distances less than 100 m, it is not correct to assume that repeat measurements will remove bias in the calculated average emission. It is currently unclear if mobile, in-situ measurements in and across the plume, even at distances shorter than 100m, would give much better results.

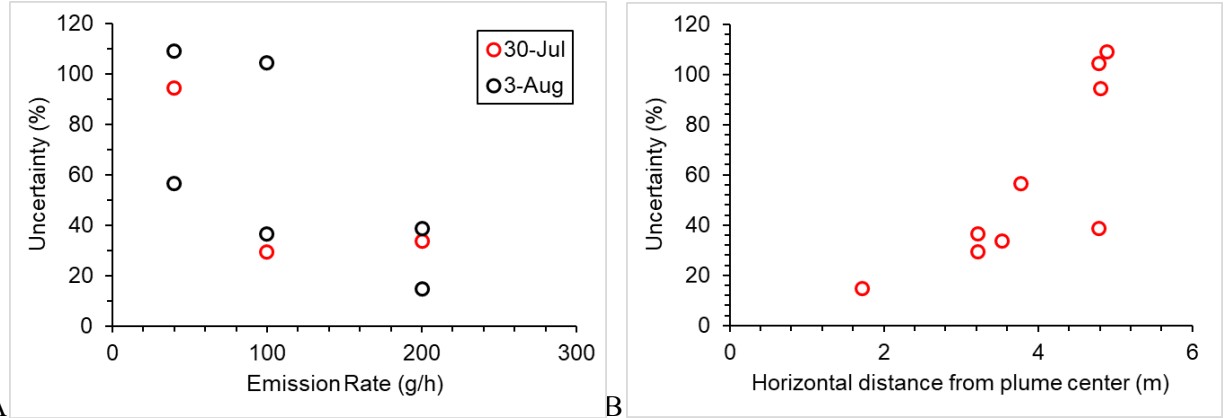

**Figure 3 A) Individual uncertainty in Gaussian Plume measurements at 40, 100 and 200 g CH$_4$ h$^{-1}$ and B) Individual uncertainties plotted against the horizontal distance from the plume center (m)**

It is also important to note that the study performed here did not simulate or account for issues which increase error in field conditions. For example, when using downwind methods (GP or bLs), the scientist may not know the exact location of the emission point and may be further downwind of the emission location. These knowledge errors may result in uncertainties, or bias in excess of what is presented here; our study should be viewed as a best case bound on the accuracy of the methods.

**5 Conclusions**

We find that both the dynamic chamber ($A_r$ = -10%, -8%, -10% at emission rates of 40, 100 and 200 g CH$_4$ h$^{-1}$, respectively) and Hi Flow ($A_r$ = -18%, -16%, -18%) repeatedly underestimate the emission, but the dynamic chamber had better accuracy. The standard deviation of emissions from these direct measurement methods remained relatively constant for emissions between 40 and 200 g CH$_4$ h$^{-1}$. The static chamber data were not presented because of safety concerns during the experiments. For the far field methods, the bLs method generally underestimated emissions ($A_r$ = +6%, -6%, -7%) while the GP method significantly overestimated the emissions ($A_r$ = +86%, +57%, +29%) despite using the same meteorological and concentration data as input. Variability in wind speed, wind direction and atmospheric stability over the 20-minute averaging period are likely to propagate through to large variability in the emission estimate, making these methods less precise than the direct measurement methods. Our results provide evidence to justify the selection of methods used to quantify emissions from abandoned oil and gas infrastructure on the basis of accuracy and precision as well as practical and economic considerations.

**Data availability**

All experimental data are presented in Supplementary Material Sections 2 and 3. Any further information can be obtained by contacting Stuart Riddick (Stuart.Riddick@colostate.edu).

**Author Contributions**

*Stuart N. Riddick:* Conceptualization, Investigation, Methodology, Supervision, Writing – original draft preparation, review and editing

*Riley Ancona*: Investigation, Data Curation and Analysis, Writing: original draft preparation

*Clay Bell:* Writing – Analysis, review and editing

*Mercy Mbua:* Writing – Analysis, review and editing

*Aidan Duggan:* Investigation

*Tim Vaughn*: Investigation, Methodology

*Kristine Bennett:* Writing – Analysis, review and editing

*Dan Zimmerle:* Writing – Analysis, review and editing

**Disclosure Statement**

The authors declare that no financial interest or benefit that has arisen from the direct applications of this research.

**Competing Interests**

The authors declare that they have no conflict of interest.

**Acknowledgements**

This work was funded by the METEC Industry Advisory Board (IAB) at Colorado State University.

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
