# Peer review of "A quantitative comparison of methods used to measure smaller methane emissions typically observed from superannuated oil and gas infrastructure"

_Atmospheric Measurement Techniques, 2022_

## Author Comment (AC1)

Reviewer 1

Title: Quantitative comparison of methods used to estimate methane emissions from small point sources

The Powerhouse Energy Campus

Colorado State University

430 North College Avenue

Fort Collins, CO 80524

E-mail: Stuart.Riddick@colostate.edu

26th April 2022

Dear Dr Chen,

We thank reviewer 1 for their comments. As suggested, we have amended the manuscript to address the reviewers' comments and have indicated changes to the manuscript in red text.

Please find our detailed responses below.

Yours sincerely,

Stuart Riddick

**Reviewer 1 General comment 1:**

One concern with this paper is that it's unclear who this work is applicable to. For example, for greenhouse gas emission inventories, there is always the emission factor approach, in contrast to what's shown in Figure 4. The authors mention oil and gas wells but without providing much context into the type of oil and gas wells, which can explain how the flow rates studied were chosen.

Response to reviewer:

The aim of this work to produce the first study to compare emissions quantified by a range of methods against known methane release rates. The target audience is anyone who has thought of, or who has used one of the methods included. As mentioned in the original introduction, these methods were chosen as they have been used extensively without a clear understanding of the precision and accuracy of quantification. Furthermore, as the methods have been used to guide policy decisions, it is important to regulators to understand how well these methods perform under controlled conditions. Therefore, we suggest this work is of interest to those currently quantifying methane emissions, those that plan to quantify methane emission and regulators that have justify decisions on policy. We also add the caveat that we only endorse measurement methodologies can be done safely wearing PPE as regulated at METEC (steel toe boot, FR overalls, hard hat, safety glasses and 4 gas monitor).

Changes to the manuscript:

In response to the reviewer's comment we have edited the opening paragraph to make the target audience more obvious.

At L27:

"Methane ($CH_4$) gas is a powerful greenhouse gas with a greenhouse warning potential 84 times larger than carbon dioxide over 100 years. Quantification of $CH_4$ emissions from abandoned wells has recently become an area of interest as studies suggest over 200 Gg $CH_4$ yr$^{-1}$ is emitted from 2.2 million abandoned wells in the US alone (US EPA, 2021). Quantifying and then plugging these wells makes them an attractive target for achieving goals set out in the Paris Agreement (Nisbet et al., 2020). Additionally, private companies are beginning initiatives to generate revenue through carbon credits gained by plugging wells and accurate quantification is essential for realizing the capital.

As there are millions of abandoned wells globally, there is a growing need to measure as many wells as quickly as possible to identify the most emissive wells. Typically, an emission from an abandoned well can be considered as an above-ground point source that is relatively small in emission size, up to 180 g $CH_4$ hour$^{-1}$ (Riddick et al., 2019a; Pekney et al., 2018; Townsend-Small et al., 2016; Boothroyd et al., 2016). Other emission sources, such as emissions pipeline leakage, are fundamentally different in behavior, where gas travels through the soil and forms an area emission at the surface, these sources require different methods for estimating the emission, e.g. mass balance or eddy covariance. Area emissions could form if a plugged well leaks from corrosion of the borehole casing, but this will not be discussed in this study.

Several methods are being used to measure emissions from these smaller point sources, i.e. less than 180 g CH$_4$ hour$^{-1}$. The chosen measurement approach depends on how close an observer can get to the source, instrumentation availability and the meteorological/micrometeorological conditions at the measurement site. Measurement methods can be classed as direct, i.e. touching/enclosing the source, and downwind measurements where access is not possible. Direct methods include static chambers (Livingston and Hutchinson, 1995), dynamic flux chambers (Riddick et al., 2019a, 2020b; Aneja et al., 2006) and Hi Flow sampling (Pekney et al., 2018; Allen et al., 2013; Brantley et al., 2015). While downwind methods include Gaussian-based plume models (Baillie et al., 2019; Caulton et al., 2014; Riddick et al., 2019b, 2020a; Edie et al., 2020; Bell et al., 2017) and Lagrangian dispersion models (Riddick et al., 2019b, 2017; Denmead, 2008; Flesch et al., 1995). Emissions calculated using the majority of these methods have not been comprehensively compared using to controlled emission source rates.

Other quantification methods are generally unsuitable for measuring emissions from abandoned wells. Infra-red cameras, such as FLIR cameras, cannot be used to quantify emission and have difficultly detecting plume smaller emissions (Zimmerle et al., 2020). Mass balance approaches are unlikely to detect the small and narrow plume from the abandoned well. Tracer release is technically demanding, takes a long time to make a single measurement and requires road access for measurement. Remote sensing has typical detection limits of 10+ kg CH$_4$ h$^{-1}$ for aircraft (Duren et al., 2019), 100+ kg CH$_4$ h$^{-1}$ for satellites (Cooper et al., 2022) and unsuitable for these types of emission source. As such, these other quantification methods will not be investigated in this study.

In general, as access becomes more restricted, emission rates larger, or safety concerns increase (such as the co-emission of harmful gases), the method used to estimate the CH$_4$ emission rate of a source must be carefully considered. From experience and the response of a 4-gas monitor, working close enough to measure emissions greater than 200 g CH$_4$ h$^{-1}$ for many of these methods (especially the chambers and Hi Flow) can be unsafe, therefore this study is limited to quantifying CH$_4$ emissions between the lowest flow METEC can produce (40 g CH$_4$ h$^{-1}$) and the highest flow we feel comfortable measuring with these methods (200 g CH$_4$ h$^{-1}$). Putting these emission ranges into real-word context, the maximum emission from unplugged and abandoned wells was measured at 177 g CH$_4$ h$^{-1}$ in West Virginia (Riddick et al., 2019a), 175 g CH$_4$ h$^{-1}$ in Pennsylvania (Pekney et al., 2018), 146 g CH$_4$ h$^{-1}$ across the US (Townsend-Small et al., 2016) and 35 g CH$_4$ h$^{-1}$ in the UK (Boothroyd et al., 2016). As most of the methods presented here require access to the source, we considered 200 g CH$_4$ h$^{-1}$ to be a sensible limit to the emission rate and is larger than the emissions observed by many previous studies. Therefore, the scope of this study is limited to estimating CH$_4$ emissions from a single point source that we would realistically be able to approach and measure, i.e. between 40 and 200 g CH$_4$ h$^{-1}$."

At L74:

**Reviewer 1 General comment 2:**

Also, the authors caution against doing measurements at sites with hydrogen sulfide and aromatic hydrocarbons, which would bias samples for regional or national inventories. From a policy perspective, wells that emit hydrogen sulfide or aromatic hydrocarbons are often prioritized for mitigation, and it would be unfortunate if we can't quantify the methane emissions being reduced through these efforts.

Response to reviewer:

We do not caution against measurement of $H_2S$ and aromatic hydrocarbons, but caution against using a static chamber to measure methane emissions. The stated aim of this study is to present controlled release data for methods that can be used to measure methane emissions from a point source between 40 and 200 g $CH_4$ $h^{-1}$. This paper only refers to the measurement of methane and does not comment on the measurement of other gases.

Change to the manuscript

At L74:

"We add the caveat that we will only present data from measurement methodologies can be conducted safely, wearing PPE as regulated at the Colorado State University Methane Emissions Technology Evaluation Center (METEC) facility in Fort Collins, CO, USA (steel toe boot, FR overalls, hard hat, safety glasses and 4-gas monitor)."

**Reviewer 1 General comment 3:**

The methods can give very different uncertainties depending on how the experiment is conducted. For example, there are many ways to implement the Gaussian Plume method, including how and where methane is analyzed and the micrometeorology measured. The same goes for the static and dynamic chambers. Overall, the authors need to add more detail on the methodology and experimental conditions (dates/times, exact equipment and supplies, etc.).

Response to reviewer:

We have added text to the manuscript (as suggested by the reviewer below) and dates/times of the experiments to the SI to clarify some of the details of the methodologies. However, the assertion that method uncertainty changes over time is conjecture. This may be true about downwind measurements, where atmospheric stability makes calculating emissions more complex resulting in propagated uncertainty, but chamber measurement uncertainties are unlikely to change over time. To address these concerns we have added text to the discussion.

Change to the manuscript

At L317:

[revised manuscript text omitted]

Below are some additional detailed comments:

**Reviewer 1 Specific comment 2:**

line 2: what is considered a "small point source"?

Response to reviewer:

In general, a point source is considered to be an emission from an aperture. In this case, the hole was a 6mm diameter tube. We have included this detail in the methods section.

Changes to the manuscript:

At L86:

"(diameter 6 mm)"

**Reviewer 1 Specific comment 2:**

line 7: how is a point source defined? At what scale?

Response to reviewer:

A point source is considered to be an emission from an aperture. In this case, the hole was a 6mm diameter tube. We have included this detail in the methods section.

Changes to the manuscript:

At L86:

"(diameter 6 mm)"

**Reviewer 1 Specific comment 3:**

line 13: not clear if static chambers are tested in this study.

Response to reviewer:

At L256 of the manuscript we describe one of the shortcomings of the static chamber as being inherently dangerous and "As such, we have not presented our measurement data and strongly encourage the use of an alternative method.".

Measurements were taken but we chose not to release the data. Instead, we included a statement that we encourage the use of any other method. To clarify this we have added a caveat to the aims of the study.

At L74:

"We add the caveat that we will only present data from measurement methodologies can be conducted safely, wearing PPE as regulated at the Colorado State University Methane Emissions Technology Evaluation Center (METEC) facility in Fort Collins, CO, USA (steel toe boot, FR overalls, hard hat, safety glasses and 4-gas monitor)."

**Reviewer 1 Specific comment 3:**

line 16: why only 200 g/h? confusing because in the text, there are three flow rates mentioned.

Response to reviewer:

Corrected as suggested.

Changes to the manuscript:

At L16:

"40 to 200 g CH$_4$ h$^{-1}$"

**Reviewer 1 Specific comment 4:**

line 40: "very small" sounds arbitrary. where does the 0.6 mg/hr come from? Kang et al. (2014) also measured negative emission rates. There also are a wide range of published measurements using this approach for much smaller fluxes found in natural environments.

Response to reviewer:

As suggested, we have edited the static chamber paragraph to remove the emission data.

Changes to the manuscript:

At L93:

"The static chambers method is relatively simple, where a container of a known volume ($V$, m$^3$) is placed over the emission source and the change in concentration ($C$, g m$^{-3}$) inside the container over time ($t$, s) can be used to calculate the emission ($Q$, g s$^{-1}$; Equation 1). The static chamber method requires no power, apart from batteries to run a fan in the chamber and is very portable. The major shortcoming of this method is that large emission sources can result in the concentration inside the chamber exceeding the CH$_4$ lower explosive limit (LEL)."

**Reviewer 1 Specific comment 5:**

line 41: replace "place" with "placed"

Response to reviewer:

Corrected, as suggested.

**Reviewer 1 Specific comment 6:**

line 43: The static chamber method does not require a gas chromatograph. In El Hachem and Kang (2022) published in Science of the Total Environment, they do not use a gas chromatograph.

Response to reviewer:

As suggested, we have removed the GC comment and added using other sensors to the discussion.

At L93:

"The static chambers method is relatively simple, where a container of a known volume ($V$, m$^3$) is placed over the emission source and the change in concentration ($C$, g m$^{-3}$) inside the container over time ($t$, s) can be used to calculate the emission ($Q$, g s$^{-1}$; Equation 1). The static chamber method requires no power, apart from batteries to run a fan in the chamber and is very portable. The major shortcoming of this method is that large emission sources can result in the concentration inside the chamber exceeding the CH$_4$ lower explosive limit (LEL)."

At L235

"To address the first shortcoming, a trace gas analyzer could be used to measure the concentrations inside the chamber. As trace gas analyzers use a pump to draw air into the measurement cavity, the analyzer could be arranged in one of two ways. Both introduce additional uncertainty into the quantification. If the gas is removed from the

chamber (i.e. the analyzer outlet is vented outside the chamber), the static chamber becomes a dynamic chamber and the analyzer flow rate must be accounted for in the quantification. If the measured gas is reintroduced to the chamber (i.e. the analyzer outlet is vented back to the chamber), a gas of lower concentration is being continually added to the "closed" system and it is therefore unclear how much uncertainty is caused by this cycling. Furthermore, the linear response of a portable trace gas analyzer, e.g. the ABB GLA131-GGA Greenhouse Gas Analyzer (https://new.abb.com/), is 100 ppm. Using the lowest emission rate in the study, 40 g $CH_4$ $h^{-1}$, and the largest chamber, 0.5 $m^3$, the concentration inside the chamber will exceed the linear range within 7 seconds. It is unlikely that gas will mix entirely throughout the chamber in 7 seconds and emission estimates are unlikely to be accurate. Another alternative could be using a lower precision sensor with a larger detection range, such as the SGX INIR- ME100 (https://sgx.cdistore.com/) that can measure from 200 ppm to 100% methane bv, but safety issues remain."

**Reviewer 1 Specific comment 7:**

line 50: what is the source of this air? Is it background air?

Response to reviewer:

Yes, this is background air that was also sampled using a GC. A description of the sampling of air inside and outside the chamber has been added to the manuscript.

Changes to the manuscript:

At L132:

"When the chamber reached steady state, three air samples were taken from inside the chamber. A background air sample was taken outside the chamber as the chamber approached steady state. The methane concentration in all air samples was measured using a gas chromatography."

**Reviewer 1 Specific comment 8:**

line 53: what is the background methane concentrations in the air? And what air is the authors referring to?

Response to reviewer:

A sample of air was taken outside the chamber as the it approached steady state. This is the sample of background air that was measured and used in the calculation.

Changes to the manuscript:

At L132:

"When the chamber reached steady state, three air samples were taken from inside the chamber. A background air sample was taken outside the chamber as the chamber approached steady state. The methane concentration in all air samples was measured using a gas chromatography."

**Reviewer 1 Specific comment 9:**

line 56: how much power is required? What type of power source is needed?

Response to reviewer:

In this case, 120 V mains power was used, however, this could be anything supplying a 12 V.

Changes to the manuscript:

At L141:

"As the experiment was conducted at METEC, 120 V mains power was used, however, in a remote location power can be supplied by anything capable of delivering a stable 12 V output."

**Reviewer 1 Specific comment 10:**

line 62: what is the current commercial HiFlow sampler? I see in the next lines that you mention the Bacharach. But I've heard that it's been discontinued. Are there others that are currently commercially available? In the previous sentence, the authors write "typical rates are 300 l/min" but that implies there are multiple types of samplers. If there is just one, why not just report the on high flow rate?

Response to reviewer:

This refers to the industry standard Bacharach HiFlow, even though the instrument's production has been discontinued. The Bacharach HiFlow has multiple flow rates and varies throughout the emission quantification, hence the inexact value. To remain instrument agnostic we have removed this information from this section.

Changes to the manuscript:

At L149:

"A Hi Flow sampler draws high volumes of air into a measurement chamber, where the concentration of $CH_4$ in the air is measured and the emission rate calculated (Equation 3). The Bacharach Hi Flow Sampler (Heath Consultants Inc., www.heathus.com) is the only current industry standard Hi Flow sampler, it draws air at between 226 and 297 l $min^{-1}$ and can measure $CH_4$ emissions between 50 g $CH_4$ $h^{-1}$ to 9 kg $CH_4$ $h^{-1}$ to an accuracy of ± 10% (Connolly et al.,

2019). A recent study commissioned by the California Air Resources Board developed open-source architecture for a new Hi Flow unit which is capable of replacing the current Bacharach Hi Flow Sampler (Vaughn et al., 2022)."

**Reviewer 1 Specific comment 11:**

line 87: the inputs to the bLS model appears to be the same as the GP model? What are the exact meteorology and micrometeorology parameters needed for the bLS model?

Response to reviewer:

The inputs used here were wind speed, wind direction, Pasquill-Gifford stability class and roughness length, as presented at L216. The roughness length was set at 2.3 cm to represent the short grass of the fetch.

Changes to the manuscript:

At L220:

"The roughness length was set at 2.3 cm to represent the short grass of the fetch. Again, it assumed that the experiments are conducted as close as possible to the source without direct access to the emission point."

**Reviewer 1 Specific comment 12:**

line 87: where is this gas concentration taken?

Response to reviewer:

The height of the measurement, 1.5 m, has been included at L191.

Changes to the manuscript:

At L191:

"1.5 m above ground level,"

**Reviewer 1 Specific comment 13:**

line 94: isn't complex topography and buildings also an issue for the GP model?

Response to reviewer:

This isn't an input to the GP model, therefore not an issue.

Changes to the manuscript:

At L198:

"Complex topography, such as building and trees, are not parameterized or accounted for by the GP model."

**Reviewer 1 Specific comment 14:**

line 96: what kind of micrometeorology data is needed?

Response to reviewer:

The inputs used here were Pasquill-Gifford stability class and roughness length, as presented at L220. The roughness length was set at 2.3 cm to represent the short grass of the fetch.

Changes to the manuscript:

At L220:

"The roughness length was set at 2.3 cm to represent the short grass of the fetch."

**Reviewer 1 Specific comment 15:**

line 100: for higher emission rates, wouldn't it be easier to do downwind measurements such that site access is less of a concern?

Response to reviewer:

This sentence is misleading and has been reworded.

Changes to manuscript

At L58:

"In general, as access becomes more restricted, emission rates larger, or safety concerns increase (such as the co-emission of harmful gases), the method used to estimate the $CH_4$ emission rate of a source must be carefully considered."

**Reviewer 1 Specific comment 16:**

line 100: what are the safety concerns here? just explosion risk due to high methane concentrations? What about H2S (See El Hachem and Kang, 2022)?

Response to reviewer:

A significant safety concern for abandoned oil and gas would be the exposure to other co-emitted gases. The measurement of $H_2S$ seems particularly hazardous without adequate PPE, even at low concentrations, 10 ppm, $H_2S$ can cause respiratory failure. However, in this study we focus solely on quantifying methane emissions using typical PPE (FRs and 4 gas monitors) and do not comment on quantifying other gases. It is financially and logistically unrealistic to expect a team measuring many sites (many groups are aiming to measure 100s of sites) to put on self-contained breathing apparatus for each measurement when a safer alternative is available.

Changes to the manuscript:

At L 55:

"In general, as access becomes more restricted, emission rates larger, or safety concerns increase (such as the co-emission of harmful gases), the method used to estimate the $CH_4$ emission rate of a source must be carefully considered. From experience and the response of a 4-gas monitor, working close enough to measure emissions greater than 200 g $CH_4$ $h^{-1}$ for many of these methods (especially the chambers and Hi Flow) can be unsafe, therefore this study is limited to quantifying $CH_4$ emissions between the lowest flow METEC can produce (40 g $CH_4$ $h^{-1}$) and the highest flow we feel comfortable measuring with these methods (200 g $CH_4$ $h^{-1}$). Putting these emission ranges into real-word context, the maximum emission from unplugged and abandoned wells was measured at 177 g $CH_4$ $h^{-1}$ in West Virginia (Riddick et al., 2019a), 175 g $CH_4$ $h^{-1}$ in Pennsylvania (Pekney et al., 2018), 146 g $CH_4$ $h^{-1}$ across the US (Townsend-Small et al., 2016) and 35 g $CH_4$ $h^{-1}$ in the UK (Boothroyd et al., 2016). As most of the methods presented here require access to the source, we considered 200 g $CH_4$ $h^{-1}$ to be a sensible limit to the emission rate and is larger than the emissions observed by many previous studies. Therefore, the scope of this study is limited to estimating $CH_4$ emissions from a single point source that we would realistically be able to approach and measure, i.e. between 40 and 200 g $CH_4$ $h^{-1}$."

At L71:

"We add the caveat that we will only present data from measurement methodologies can be conducted safely, wearing PPE as regulated at the Colorado State University Methane Emissions Technology Evaluation Center (METEC) facility in Fort Collins, CO, USA (steel toe boot, FR overalls, hard hat, safety glasses and 4-gas monitor)."

**Reviewer 1 Specific comment 17:**

line 102: what is meant by "able to approach"? How close to the single point source in meters?

Response to reviewer:

From experience, it becomes difficult to work effectively in areas near NG emissions greater than 200 g $CH_4$ $h^{-1}$. As a rule of thumb at this emission rate, we suggest access to areas closer than 10 m of the source be restricted. As most of the methods require access to the source, we considered this to be a sensible limit to emission.

Changes to the manuscript:

At L55

"In general, as access becomes more restricted, emission rates larger, or safety concerns increase (such as the co-emission of harmful gases), the method used to estimate the $CH_4$ emission rate of a source must be carefully considered. From experience and the response of a 4-gas monitor, working close enough to measure emissions greater than 200 g $CH_4$ $h^{-1}$ for many of these methods (especially the chambers and Hi Flow) can be unsafe, therefore this study is limited to quantifying $CH_4$ emissions between the lowest flow METEC can produce (40 g $CH_4$ $h^{-1}$) and the highest flow we feel comfortable measuring with these methods (200 g $CH_4$ $h^{-1}$). Putting these emission ranges into real-word context, the maximum emission from unplugged and abandoned wells was measured at 177 g $CH_4$ $h^{-1}$ in West Virginia (Riddick et al., 2019a), 175 g $CH_4$ $h^{-1}$ in Pennsylvania (Pekney et al., 2018), 146 g $CH_4$ $h^{-1}$ across the US (Townsend-Small et al., 2016) and 35 g $CH_4$ $h^{-1}$ in the UK (Boothroyd et al., 2016). As most of the methods presented here require access to the source, we considered 200 g $CH_4$ $h^{-1}$ to be a sensible limit to the emission rate and is larger than the emissions observed by many previous studies. Therefore, the scope of this study is limited to estimating $CH_4$ emissions from a single point source that we would realistically be able to approach and measure, i.e. between 40 and 200 g $CH_4$ $h^{-1}$."

At L74:

"We add the caveat that we will only present data from measurement methodologies can be conducted safely, wearing PPE as regulated at the Colorado State University Methane Emissions Technology Evaluation Center (METEC) facility in Fort Collins, CO, USA (steel toe boot, FR overalls, hard hat, safety glasses and 4-gas monitor)."

**Reviewer 1 Specific comment 18:**

line 103: what are the emission rates considered? In the abstract, it was only for 200 g/h but there are three mentioned later. Need to be consistent throughout.

Response to reviewer:

Have amended to bracket the emissions throughout the manuscript.

**Reviewer 1 Specific comment 19:**

line 108: why the cut off at 200 g/h? There should be a paragraph on the literature for tests at >200 g/h and describe why those studies are not applicable here.

Response to reviewer:

This was a health and safety regulation at METEC. From experience, it becomes difficult to work effectively in areas near NG emissions greater than 200 g $CH_4$ $h^{-1}$. As a rule of thumb at this emission rate, we suggest access to areas closer than 10 m of the source be restricted. As most of the methods require access to the source, we considered this to be a sensible limit to emission. This upper limit is also larger than most of the emissions observed from abandoned oil and gas wells and emissions have been presented in the text.

Changes to the manuscript:

At L56:

"From experience and the response of a 4-gas monitor, working close enough to measure emissions greater than 200 g $CH_4$ $h^{-1}$ for many of these methods (especially the chambers and Hi Flow) can be unsafe, therefore this study is limited to quantifying $CH_4$ emissions between the lowest flow METEC can produce (40 g $CH_4$ $h^{-1}$) and the highest flow we feel comfortable measuring with these methods (200 g $CH_4$ $h^{-1}$). Putting these emission ranges into real-word context, the maximum emission from unplugged and abandoned wells was measured at 177 g $CH_4$ $h^{-1}$ in West Virginia (Riddick et al., 2019a), 175 g $CH_4$ $h^{-1}$ in Pennsylvania (Pekney et al., 2018), 146 g $CH_4$ $h^{-1}$ across the US (Townsend-Small et al., 2016) and 35 g $CH_4$ $h^{-1}$ in the UK (Boothroyd et al., 2016). As most of the methods presented here require access to the source, we considered 200 g $CH_4$ $h^{-1}$ to be a sensible limit to the emission rate and is larger than the emissions observed by many previous studies. Therefore, the scope of this study is limited to estimating $CH_4$ emissions from a single point source that we would realistically be able to approach and measure, i.e. between 40 and 200 g $CH_4$ $h^{-1}$."

At L74:

"We add the caveat that we will only present data from measurement methodologies can be conducted safely, wearing PPE as regulated at the Colorado State University Methane Emissions Technology Evaluation Center (METEC) facility in Fort Collins, CO, USA (steel toe boot, FR overalls, hard hat, safety glasses and 4-gas monitor)."

**Reviewer 1 Specific comment 20:**

line 110: the exact dates and times in which each experiment took place needs to be provided.

Response to reviewer:

Have been added as Supplementary Material Section 4

**Reviewer 1 Specific comment 21:**

line 111: it's unclear which exact experiments were done. it would be helpful if the authors could provide a spreadsheet with all the tests that were conducted in the supporting information.

Response to reviewer:

Have been added as Supplementary Material Section 4

**Reviewer 1 Specific comment 22:**

line 115: how were the emission rates set? What is the type of flow controller? What are the gases that are used?

Response to reviewer:

At the METEC site, compressed natural gas, with methane compositions ranging from 85 to 95%vol, is supplied from two 145 L cylinders. Flown rates are controlled using a pressure regulator and precision orifices.

Changes to the manuscript:

At L82

"At the METEC site, compressed natural gas, with methane compositions ranging from 85 to 95%vol, is supplied from two 145 L cylinders and flow rates controlled using a pressure regulator and precision orifices."

**Reviewer 1 Specific comment 23:**

line 115-116: how do you define/determine what is safe or not?

Response to reviewer:

From experience, it becomes difficult to work effectively in areas near NG emissions greater than 200 g $CH_4\,h^{-1}$. As a rule of thumb at this emission rate, we suggest access to areas closer than 10 m of the source be restricted this is often evident from the response of the 4-gas monitor. As most of the methods require access to the source, we considered this to be a sensible limit to emission. The 4-gas monitor indicated low oxygen while conducting the static chamber measurements and the measurements we halted.

Changes to the manuscript:

At L56:

"From experience and the response of a 4-gas monitor, working close enough to measure emissions greater than 200 g $CH_4\ h^{-1}$ for many of these methods (especially the chambers and Hi Flow) can be unsafe, therefore this study

is limited to quantifying $CH_4$ emissions between the lowest flow METEC can produce (40 g $CH_4$ $h^{-1}$) and the highest flow we feel comfortable measuring with these methods (200 g $CH_4$ $h^{-1}$)."

At L74:

"We add the caveat that we will only present data from measurement methodologies can be conducted safely, wearing PPE as regulated at the Colorado State University Methane Emissions Technology Evaluation Center (METEC) facility in Fort Collins, CO, USA (steel toe boot, FR overalls, hard hat, safety glasses and 4-gas monitor)."

**Reviewer 1 Specific comment 24:**

line 116: why not lower than 40 g/hr?

Response to reviewer:

The lowest emission rate METEC can produce is 40 g/hr.

**Reviewer 1 Specific comment 25:**

line 125: it's unclear if a static chamber measurement was done?

Response to reviewer:

The static chamber measurements were not reported.  We felt the method was intrinsically unsafe when other methods were safer.  During the experiment the 4-gas monitor on our researcher indicated low oxygen while conducting the static chamber measurements and the measurements we halted.

Changes to the manuscript:

At L74:

"We add the caveat that we will only present data from measurement methodologies can be conducted safely, wearing PPE as regulated at the Colorado State University Methane Emissions Technology Evaluation Center (METEC) facility in Fort Collins, CO, USA (steel toe boot, FR overalls, hard hat, safety glasses and 4-gas monitor)."

**Reviewer 1 Specific comment 26:**

line 127: what is the location and size of the fan inside the chamber selected? and how was this selected?

Response to reviewer:

We used a fan similar to the one presented in Riddick et al. (2019). This was selected to ensure that the air was fully mixed throughout the cylindrical chamber.

Changes to the manuscript:

At L102:

"A fan was secured inside the chamber and used to circulate the air following the method of (Riddick et al. 2019a) to ensure the air inside the chamber was fully mixed."

**Reviewer 1 Specific comment 27:**

line 128: What is meant by "three further air samples? Further to what?

Response to reviewer:

Should have said that a sample was taken at time $t_0$.

Changes to the manuscript:

L 106:

"When the chamber is sealed with the ground, following Riddick et al. (2019a), an air sample is drawn using a gas syringe. During the experiment at least three further air samples are taken at regular intervals (Pihlatie et al., 2013; Collier et al., 2014),"

**Reviewer 1 Specific comment 28:**

line 132: what where the shapes of the chambers? what is the aspect ratio (height to diameter)?

Response to reviewer:

These were all rigid plastic cylindrical chambers, with heights approximately 1.5 times the diameter.

Changes to the manuscript:

At L107

"Two sizes of static chambers were used in this experiment (0.12 $m^3$ and 0.5 $m^3$; Figure 1), the chambers were made from rigid plastic cylindrical chambers, with heights approximately 1.5 times the chamber's diameter. The chambers sizes was based on a measurable concentration change over time for given release rates, however, it is unlikely that the larger size is practical for field deployment."

**Reviewer 1 Specific comment 29:**

line 133: how was the quality of the ground seal determined?

Response to reviewer:

Seal was made in line with the information in Riddick et al. (2019).

Changes to the manuscript:

L106

", following Riddick et al. (2019a),"

**Reviewer 1 Specific comment 30:**

line 135: is this experiment a copy of Kang et al (2014), Pihalatie et al (2013), or Collier et al (2014)? Which one took four samples? Kang et al (2014) took 7 to 8 samples.

Response to reviewer:

As stated in L107 "During the experiment, four sample of 25 ml of air were drawn from the chamber using a 50 ml gas syringe at equal time intervals (Pihlatie et al., 2013; Collier et al., 2014)".

**Reviewer 1 Specific comment 31:**

line 135: there is a "s" missing. it should be "four samples".

Response to reviewer:

Corrected

**Reviewer 1 Specific comment 32:**

line 137: were there duplicates and blanks taken?

Response to reviewer:

Blanks (field or lab) were not required as all air samples were measured on the day of the experiment (L171 should say hours not days). Also, all experiments were repeated 3 times.

Changes to the manuscript:

At L118

"All samples were measured within two hours of collection.  All experiments were repeated three times."

**Reviewer 1 Specific comment 33:**

line 142: does this imply that emission rates were calculated for test even if only three samples were collected?

Response to reviewer:

Emission rates were calculated from the four measurements.

**Reviewer 1 Specific comment 34:**

line 146: why was this chamber size selected? what is the shape and aspect ratio? How useful is this size for field measurements of oil and gas wells?

Response to reviewer:

These chambers sizes were chosen as they are what we had to hand at METEC and calculated as a safe size to use. These sizes are impractical for field deployment.

Changes to the manuscript:

L107:

"Two sizes of static chambers were used in this experiment (0.12 $m^3$ and 0.5 $m^3$; Figure 1), the chambers were made from rigid plastic cylindrical chambers, with heights approximately 1.5 times the chamber's diameter.  The chambers sizes was based on a measurable concentration change over time for given release rates, however, it is unlikely that the larger size is practical for field deployment."

**Reviewer 1 Specific comment 35:**

line 146: what type of plastic is used?

Response to reviewer:

Rigid plastic

Changes to the manuscript:

At L110

"rigid plastic"

**Reviewer 1 Specific comment 36:**

line 149: what is the detection limit of the HXG-2D? What are the methane concentrations observed inside the chamber?

Response to reviewer:

Added to the manuscript

Changes to the manuscript:

At L122:

"detection limit 10 ppm and range 0 to 40,000 ppm"

**Reviewer 1 Specific comment 37:**

line 150-151: blanks and duplicates taken?

Response to reviewer:

Blanks (field or lab) were not required as all air samples were measured on the day of the experiment (L186 should say hours not days). Also, all experiments were repeated 3 times.

Changes to the manuscript:

At L148

"As with the samples from the static chamber, the vials were measured within two hours of collection. All experiments were repeated three times."

**Reviewer 1 Specific comment 38:**

line 155: how big is the hose end? Was the source enclosed by the Hi-Flow sampler?

Response to reviewer:

The Hi Flow has a bag, this was placed over the leak.

Changes to the manuscript:

At L162:

"the bag containing the hose"

**Reviewer 1 Specific comment 39:**

line 167: any concerns with topography and large objects (e.g., buildings, trees, and other infrastructure)?

Response to reviewer:

No, METEC is a big open flat site.

**Reviewer 1 Specific comment 40:**

line 177: what is the minimum distance between the sources and the detector?

Response to reviewer:

5 m, as stated on Line 191

**Reviewer 1 Specific comment 41:**

line 181: how different were the environments in which the experiments were conducted? Importantly, were experiments described in sections 2.1 to 2.5 all conducted on the same day. If they were all conducted on different days, then the uncertainties calculated cannot be directly compared.

Response to reviewer:

This study is the first to test the uncertainty of each of these methods in a controlled test. Therefore, there is no evidence to suggest that the uncertainty will change on different days in different environmental conditions.

**Reviewer 1 Specific comment 42:**

line 190-191: The collection of gas vials is not a requirement of the static chamber methodology.

Response to reviewer:

This is true, but this is the method we are testing.

Using a trace gas analyzer to measure the concentrations inside the chamber is questionable. If the gas is removed from the chamber (i.e. the analyzer outlet is vented outside the chamber) this system then becomes a dynamic chamber. If the measured gas is reintroduced to the chamber (i.e. the analyzer outlet is vented back to the chamber) this means gas of a lower concentration is being continually added to the "closed" system. It is unclear of the uncertainty caused by this cycling. To my knowledge, using a gas analyzer in series with the static chamber has never been tested.

Change to the manuscript:

At L239:

"To address the first shortcoming, a trace gas analyzer could be used to measure the concentrations inside the chamber. As trace gas analyzers use a pump to draw air into the measurement cavity, the analyzer could be arranged in one of two ways. Both introduce additional uncertainty into the quantification. If the gas is removed from the chamber (i.e. the analyzer outlet is vented outside the chamber), the static chamber becomes a dynamic chamber and the analyzer flow rate must be accounted for in the quantification. If the measured gas is reintroduced to the chamber (i.e. the analyzer outlet is vented back to the chamber), a gas of lower concentration is being continually added to the "closed" system and it is therefore unclear how much uncertainty is caused by this cycling. Furthermore, the linear response of a portable trace gas analyzer, e.g. the ABB GLA131-GGA Greenhouse Gas Analyzer (https://new.abb.com/), is 100 ppm. Using the lowest emission rate in the study, 40 g $CH_4$ $h^{-1}$, and the largest chamber, 0.5 $m^3$, the concentration inside the chamber will exceed the linear range within 7 seconds. It is unlikely that gas will mix entirely throughout the chamber in 7 seconds and emission estimates are unlikely to be accurate. Another alternative could be using a lower precision sensor with a larger detection range, such as the SGX INIR- ME100 (https://sgx.cdistore.com/) that can measure from 200 ppm to 100% methane bv, but safety issues remain."

At L257:

"The static chamber could be automated to release gas when $CH_4$ concentration inside the chamber approaches LEL to prevent chamber becoming explosive. The major shortcoming of this strategy is that the automation of a chamber takes away the operator's control of when gas is released, which could happen at an inconvenient during measurement. If an automated system is used for collecting gas of unknown composition self-contained breathing apparatus should be worn."

**Reviewer 1 Specific comment 43:**

line 193: the static chamber can be used with a methane analyzer (e.g., El Hachem and Kang, 2022), overcoming the first and second shortcoming.

Response to reviewer:

As the method in El Hachem and Kang, 2022 is presented, a GasScouter is used to measure the concentration of gas within a chamber. It does not comment on whether the outlet of the GasScouter is returned to the chamber, therefore, it is not clear if this is actually a static chamber.

The static chamber is inherently dangerous and we cannot advocate it's use in measuring unknown compositions of gas at unknown flow rates. It is unreasonable to assume that measurement teams would want to use self-contained breathing apparatus for every well they measure at if there are safer alternative methods available.

**Reviewer 1 Specific comment 44:**

line 196-198: El Hachem and Kang (2022) conducted measurements from H2S-emitting wells using a self-contained breathing apparatus. There are many options available in industry to ensure safe working conditions when toxic gases are present.

Response to reviewer:

It is unreasonable to expect full breathing apparatus as mandatory at all measurement locations. As such we have added a caveat to the manuscript.

Changes to the manuscript:

At L74:

"We add the caveat that we will only present data from measurement methodologies can be conducted safely wearing PPE as regulated at the Colorado State University Methane Emissions Technology Evaluation Center (METEC) facility in Fort Collins, CO, USA (steel toe boot, FR overalls, hard hat, safety glasses and 4-gas monitor)."

**Reviewer 1 Specific comment 43:**

line 198: what about for measuring low emitting sources?

Response to reviewer:

Gas composition is still unknown even for low methane emissions; therefore the risks remain. This is a major shortcoming and other safer methods should be used.

**Reviewer 1 Specific comment 44:**

line 200: why isn't "cost" italicized like the rest?

Response to reviewer:

Changed as suggested.

Reviewer 1 Specific comment 45:

Table 1: I'm surprised that the HiFlow sampler is only $5k. Is this correct? The static and dynamic chamber measurements conducted here use a GC, which is around $50k. So it's definitely not free. Even just getting the gas concentrations analyzed elsewhere is not free.

Response to reviewer:

Have added text to that effect.

Change to manuscript

At L272

◊ Cost of GC analysis will vary by laboratory.

**Reviewer 1 Specific comment 46:**

Table 1: why is there no time for measurement and analysis for the static chamber? same for the accuracy.

Response to reviewer:

As mentioned above the static chamber is not presented here as we found it inherently dangerous.

At L271

"- the static chamber data is not presented here as the method was found to be inherently dangerous."

**Reviewer 1 Specific comment 47:**

line 209: there are other ways to reduce the potential of CH4 concentrations in the chamber reaching explosive levels when using static chambers.

Response to reviewer:

Have added some text to the discussion about using automated opening chambers.

At L251:

"The static chamber could be automated to release gas when $CH_4$ concentration inside the chamber approaches LEL to prevent chamber becoming explosive.  The major shortcoming of this strategy is that the automation of a chamber takes away the operator's control of when gas is released, which could happen at an inconvenient during measurement.  If an automated system is used for collecting gas of unknown composition self-contained breathing apparatus should be worn."

**Reviewer 1 Specific comment 48:**

line 210: the need for a power source is another important shortcoming of the dynamic chamber method.

Response to reviewer:

Noted

Changes to the manuscript:

L277

"and a 12 V power source for the pump"

**Reviewer 1 Specific comment 49:**

line 266-267: what is the dynamic chamber the HiFLow more accurate than?

Response to reviewer:

This sentence doesn't make sense and has been removed.

**Reviewer 1 Specific comment 50:**

line 273: who is this decision-making paradigm for?

Response to reviewer:

On reflection we have removed the decision making paradigm.

**Reviewer 1 Specific comment 51:**

line 282: what are the conditions in this study? This needs to be better described to assess the applicability of the results.

Response to reviewer:

As described above, this study is the first to test the uncertainty of each of these methods in a controlled test. Therefore, there is no evidence to suggest that the uncertainty will change in different environmental conditions.

**Reviewer 1 Specific comment 52:**

Figure 4. Many estimates (e.g., the USEPA's GHGI) involve wells with no measurements. It's not possible for all wells to be measured. So if there is no trace gas analyzer, there is always the emission factor approach. But of course, that's not a good predictor of the emissions at a given well but over some large population, it may be representative. So this brings us back to the question of who this figure is for. This figure needs more context in the caption and the text.

Response to reviewer:

This aim of this work is to provide evidence as to how well a method performs quantifying emissions between 40 g $CH_4$ $h^{-1}$ and 200 g $CH_4$ $h^{-1}$.

As you say, emission factors do not quantify emissions from an individual source, therefore are unlikely to be representative.

We suggest this work is of interest to those currently quantifying methane emissions, those that plan to quantify methane emission and regulators that have justify decisions on policy.

---

## Author Comment (AC2)

Reviewer 2

Title: Quantitative comparison of methods used to estimate methane emissions from small point sources

The Powerhouse Energy Campus

Colorado State University

430 North College Avenue

Fort Collins, CO 80524

E-mail:  Stuart.Riddick@colostate.edu

26[th] April 2022

Dear Dr Chen,

We thank reviewer 2 for their comments.  As suggested, we have amended the manuscript to address the reviewers' comments and have indicated changes to the manuscript in red text.

Please find our detailed responses below.

Yours sincerely,

Stuart Riddick

**Reviewer 2 General comment 1:**

The state of the art in the introduction doesn't acknowledge other available techniques. The selection of methods reproduced in the study is not explicit, and the reasons for ignoring/discarding other techniques is not clarified.

Response to reviewer:

The genesis of this publication was the repeated question asked of METEC research scientists: "how good are these methods at measuring emissions typically seen from abandoned oil and gas wells and what are the benefits of repeat experiments?". Here, we present the methods most commonly used to quantify emissions and conduct blinded experiments to how representative a single quantification measurement can be and then investigate the utility of repeat measurements. Other methods that the reviewer has listed below are simply not suitable for measuring emissions typical of the majority of abandoned wells. FLIR is not a quantification approach, mass balance could be used but I have never come across it being used to measure individual point sources, tracer release is a long a technically tricky approach and ill-suited to this sort of hit-and-run measurement approach, and remote sensing has typical detection limits of 10+ kg $CH_4$ $h^{-1}$ for drones and 100+ kg $CH_4$ $h^{-1}$ for aircraft and completely unsuitable.

Changes to the manuscript:

To clarify our objective we have added text at L 27:

"Methane ($CH_4$) gas is a powerful greenhouse gas with a greenhouse warning potential 84 times larger than carbon dioxide over 100 years. Quantification of $CH_4$ emissions from abandoned wells has recently become an area of interest as studies suggest over 200 Gg $CH_4$ $yr^{-1}$ is emitted from 2.2 million abandoned wells in the US alone (US EPA, 2021). Quantifying and then plugging these wells makes them an attractive target for achieving goals set out in the Paris Agreement (Nisbet et al., 2020). Additionally, private companies are beginning initiatives to generate revenue through carbon credits gained by plugging wells and accurate quantification is essential for realizing the capital.

As there are millions of abandoned wells globally, there is a growing need to measure as many wells as quickly as possible to identify the most emissive wells. Typically, an emission from an abandoned well can be considered as an above-ground point source that is relatively small in emission size, up to 180 g $CH_4$ $hour^{-1}$ (Riddick et al., 2019a; Pekney et al., 2018; Townsend-Small et al., 2016; Boothroyd et al., 2016). Other emission sources, such as emissions pipeline leakage, are fundamentally different in behavior, where gas travels through the soil and forms an area emission at the surface, these sources require different methods for estimating the emission, e.g. mass balance or eddy covariance. Area emissions could form if a plugged well leaks from corrosion of the borehole casing, but this will not be discussed in this study.

Several methods are being used to measure emissions from these smaller point sources, i.e. less than 180 g $CH_4$ hour$^{-1}$. The chosen measurement approach depends on how close an observer can get to the source, instrumentation availability and the meteorological/micrometeorological conditions at the measurement site. Measurement methods can be classed as direct, i.e. touching/enclosing the source, and downwind measurements where access is not possible. Direct methods include static chambers (Livingston and Hutchinson, 1995), dynamic flux chambers (Riddick et al., 2019a, 2020b; Aneja et al., 2006) and Hi Flow sampling (Pekney et al., 2018; Allen et al., 2013; Brantley et al., 2015). While downwind methods include Gaussian-based plume models (Baillie et al., 2019; Caulton et al., 2014; Riddick et al., 2019b, 2020a; Edie et al., 2020; Bell et al., 2017) and Lagrangian dispersion models (Riddick et al., 2019b, 2017; Denmead, 2008; Flesch et al., 1995). Emissions calculated using the majority of these methods have not been comprehensively compared using to controlled emission source rates.

Other quantification methods are generally unsuitable for measuring emissions from abandoned wells. Infra-red cameras, such as FLIR cameras, cannot be used to quantify emission and have difficultly detecting plume smaller emissions (Zimmerle et al., 2020). Mass balance approaches are unlikely to detect the small and narrow plume from the abandoned well. Tracer release is technically demanding, takes a long time to make a single measurement and requires road access for measurement. Remote sensing has typical detection limits of 10+ kg $CH_4$ h$^{-1}$ for aircraft (Duren et al., 2019), 100+ kg $CH_4$ h$^{-1}$ for satellites (Cooper et al., 2022) and unsuitable for these types of emission source. As such, these other quantification methods will not be investigated in this study.

In general, as access becomes more restricted, emission rates larger, or safety concerns increase (such as the co-emission of harmful gases), the method used to estimate the $CH_4$ emission rate of a source must be carefully considered. From experience and the response of a 4-gas monitor, working close enough to measure emissions greater than 200 g $CH_4$ h$^{-1}$ for many of these methods (especially the chambers and Hi Flow) can be unsafe, therefore this study is limited to quantifying $CH_4$ emissions between the lowest flow METEC can produce (40 g $CH_4$ h$^{-1}$) and the highest flow we feel comfortable measuring with these methods (200 g $CH_4$ h$^{-1}$). Putting these emission ranges into real-word context, the maximum emission from unplugged and abandoned wells was measured at 177 g $CH_4$ h$^{-1}$ in West Virginia (Riddick et al., 2019a), 175 g $CH_4$ h$^{-1}$ in Pennsylvania (Pekney et al., 2018), 146 g $CH_4$ h$^{-1}$ across the US (Townsend-Small et al., 2016) and 35 g $CH_4$ h$^{-1}$ in the UK (Boothroyd et al., 2016). As most of the methods presented here require access to the source, we considered 200 g $CH_4$ h$^{-1}$ to be a sensible limit to the emission rate and is larger than the emissions observed by many previous studies. Therefore, the scope of this study is limited to estimating $CH_4$ emissions from a single point source that we would realistically be able to approach and measure, i.e. between 40 and 200 g $CH_4$ h$^{-1}$."

**Reviewer 2 General comment 2:**

There is a lack of context elaborating on the specific needs of the industry (e.g. buried sources are ignored or implicitly included as the paper seem to focus on aerial point sources), and possibly introducing some sort of statistical distribution of leak size would be useful.

Response to reviewer:

Emissions from buried oil and gas infrastructure is a different study. The authors are involved in many studies to investigate this (R-PLUME Large Pipeline Leaks - Energy Institute (colostate.edu); Upstream Pipeline Safety, Integrity and Detection - Energy Institute (colostate.edu); Innovative Sensor Network for Subsurface Emissions - Energy Institute (colostate.edu)) and, unlike above-ground point sources, below ground emissions evolve to become area emissions at the surface with methane sinks throughout the surface as well as at the surface-atmosphere interface. To our knowledge, evaluating below-surface quantification methods using controlled releases has never been conducted before and would merit a separate publication.

The data presented in this manuscript represents the first time a controlled release has been used to evaluate and compare the performance of five above-ground point-source emission quantification methods. This presents a major step-forward in understanding utility of each these quantification methods. Additionally, we present data for researchers, regulators or private operators on which method is the most suitable for deployment in a given circumstance with operational shortcomings, such as instrument availability.

Changes to the manuscript:

To highlight why we are measuring above-ground point sources we have included the following text.

At L28:

[revised manuscript text omitted]

**Reviewer 2 General comment 4:**

The "decision making paradigm" (Sect 4.2) is limited in scope by ignoring other techniques and situations that may representative of the industry, and it seems to operates in its own limited rationality, letting the reader ignore other works.

Response to reviewer:

After thinking about this we have decided to remove the flow diagram.

Specific comments

**Reviewer 2 Specific comment 1:**

L16: not only for 200 g/h.

Response to reviewer:

As suggested, have included the range of emissions.

Changes to the manuscript:

At L 16:

"for emissions of 40 to 200 g $CH_4$ $h^{-1}$"

**Reviewer 2 Specific comment 2:**

L29: why is this 200g/h threshold important? Is there a scientific rationale? Is it specifically representative of situations or technical challenges in the industry?

Response to reviewer:

From experience, it becomes difficult to work effectively in areas near NG emissions greater than 200 g $CH_4$ $h^{-1}$. As a rule of thumb at this emission rate, we suggest access to areas closer than 10 m of the source be restricted. As most of the methods require access to the source, we considered this to be a sensible limit to emission and it is more than the upper limit to emissions measurements in the field. The maximum emission from unplugged and abandoned wells was measured at 177 g $CH_4$ $h^{-1}$ in West Virginia (Riddick et al., 2019a), 175 g $CH_4$ $h^{-1}$ in Pennsylvania (Pekney et al., 2018), 146 g $CH_4$ $h^{-1}$ across the US (Townsend-Small et al., 2016) and 35 g $CH_4$ $h^{-1}$ in the UK (Boothroyd et al., 2016).

Changes to the manuscript:

At L59:

"From experience and the response of a 4-gas monitor, working close enough to measure emissions greater than 200 g $CH_4$ $h^{-1}$ for many of these methods (especially the chambers and Hi Flow) can be unsafe, therefore this study is limited to quantifying $CH_4$ emissions between the lowest flow METEC can produce (40 g $CH_4$ $h^{-1}$) and the highest

flow we feel comfortable measuring with these methods (200 g $CH_4$ $h^{-1}$). Putting these emission ranges into real-word context, the maximum emission from unplugged and abandoned wells was measured at 177 g $CH_4$ $h^{-1}$ in West Virginia (Riddick et al., 2019a), 175 g $CH_4$ $h^{-1}$ in Pennsylvania (Pekney et al., 2018), 146 g $CH_4$ $h^{-1}$ across the US (Townsend-Small et al., 2016) and 35 g $CH_4$ $h^{-1}$ in the UK (Boothroyd et al., 2016). As most of the methods presented here require access to the source, we considered 200 g $CH_4$ $h^{-1}$ to be a sensible limit to the emission rate and is larger than the emissions observed by many previous studies. Therefore, the scope of this study is limited to estimating $CH_4$ emissions from a single point source that we would realistically be able to approach and measure, i.e. between 40 and 200 g $CH_4$ $h^{-1}$."

At L74:

"We add the caveat that we will only present data from measurement methodologies can be conducted safely, wearing PPE as regulated at the Colorado State University Methane Emissions Technology Evaluation Center (METEC) facility in Fort Collins, CO, USA (steel toe boot, FR overalls, hard hat, safety glasses and 4-gas monitor)."

**Reviewer 2 Specific comment 3:**

L31-38: A number of techniques and approaches (FLIR, mass balance, tracer release, remote sensing…) are ignored in this study. Their existence and their absence here should be acknowledged and thoroughly commented. They may not be included in this study for some (presumably good) reasons?

As presented above, we have clearly stated that this is for measuring small above-ground point source emissions and particularly focusses on abandoned oil and gas wells. Other methods that the reviewer has listed below are simply not suitable for measuring emissions typical of the majority of abandoned wells. FLIR is not a quantification approach, mass balance could be used but I have never come across it being used to measure individual point sources, tracer release is a long a technically tricky approach and ill-suited to this sort of hit-and-run measurement approach, and remote sensing has typical detection limits of 10+ kg $CH_4$ $h^{-1}$ for drones and 100+ kg $CH_4$ $h^{-1}$ for aircraft and completely unsuitable.

Changes to the manuscript:

At L52:

"Other quantification methods are generally unsuitable for measuring emissions from abandoned wells. Infra-red cameras, such as FLIR cameras, cannot be used to quantify emission and have difficultly detecting plume smaller emissions (Zimmerle et al., 2020). Mass balance approaches are unlikely to detect the small and narrow plume from the abandoned well. Tracer release is technically demanding, takes a long time to make a single measurement and requires road access for measurement. Remote sensing has typical detection limits of 10+ kg $CH_4$ $h^{-1}$ for aircraft (Duren et al., 2019), 100+ kg $CH_4$ $h^{-1}$ for satellites (Cooper et al., 2022) and unsuitable for these types of emission source. As such, these other quantification methods will not be investigated in this study."

**Reviewer 2 Specific comment 4:**

L37: "despite the interest in developing methods": unclear

Response to reviewer:

Have reworded the sentence

Changes to the manuscript:

At L47:

"Emissions calculated using the majority of these methods have not been comprehensively compared using to controlled emission source rates."

**Reviewer 2 Specific comment 5:**

L78: what is "total reflection of CH4 at the surface"?

Response to reviewer:

This reflects an amendment to the original Gaussian Plume model where gas reflected from the surface of the Earth is accounted for in the downwind plume.

Changes to the manuscript:

At L177:

"where gas reflected from the surface of the Earth is accounted for in the downwind plume."

**Reviewer 2 Specific comment 6:**

L79: I would argue that mass balance is the simplest method, rather than GP.

Response to reviewer:

Fair enough, however, mass balance is generally not used for abandoned wells.

Changes to the manuscript:

At L179:

"considered here"

**Reviewer 2 Specific comment 7:**

L81: perfect "gaussian" plumes are indeed seldom met in nature. But also it is rare to have lonely 'weak' plumes in an industrial setting, so the GP approach needs to account somehow for multiple sources.

Response to reviewer:

Yes, any background interference can be accounted for up upwind and downwind measurement. The exclusion of this was an oversight in the manuscript and should have been highlighted in the text. Here we define the enhancement as the difference between the downwind concentration and the background concentration measured upwind.

Changes to the manuscript:

At L171:

"enhancement"

At L176:

"The enhancement is defined as the difference between the downwind concentration and the background concentration measured upwind."

**Reviewer 2 Specific comment 8:**

L98: is it the same 38% uncertainty applicable for 4ug/h and 3kg/h?

Response to reviewer:

This is the major shortcoming of non-controlled release emission uncertainties and doesn't account for any variability in the emission size or variability in environmental conditions. This uncertainty was calculated as a desk based study and derived from the root of the sum of the individual sources of uncertainty squared.

Changes to the manuscript:

At L47:

"Emissions calculated using the majority of these methods have not been comprehensively compared using to controlled emission source rates."

**Reviewer 2 Specific comment 9:**

L102: if the 200g/h limit is for safety/practical reason, is it still useful in real life?

From experience, it becomes difficult to work effectively in areas near NG emissions greater than 200 g $CH_4$ $h^{-1}$. As a rule of thumb at this emission rate, we suggest access to areas closer than 10 m of the source be restricted. As most of the methods require access to the source, we considered this to be a sensible limit to emission and it is more than the upper limit to emissions measurements in the field. The maximum emission from unplugged and abandoned wells was measured at 177 g $CH_4$ $h^{-1}$ in West Virginia (Riddick et al., 2019a), 175 g $CH_4$ $h^{-1}$ in

Pennsylvania (Pekney et al., 2018), 146 g $CH_4$ $h^{-1}$ across the US (Townsend-Small et al., 2016) and 35 g $CH_4$ $h^{-1}$ in the UK (Boothroyd et al., 2016).

Changes to the manuscript:

At L59:

"From experience and the response of a 4-gas monitor, working close enough to measure emissions greater than 200 g $CH_4$ $h^{-1}$ for many of these methods (especially the chambers and Hi Flow) can be unsafe, therefore this study is limited to quantifying $CH_4$ emissions between the lowest flow METEC can produce (40 g $CH_4$ $h^{-1}$) and the highest flow we feel comfortable measuring with these methods (200 g $CH_4$ $h^{-1}$).  Putting these emission ranges into real-word context, the maximum emission from unplugged and abandoned wells was measured at 177 g $CH_4$ $h^{-1}$ in West Virginia (Riddick et al., 2019a), 175 g $CH_4$ $h^{-1}$ in Pennsylvania (Pekney et al., 2018), 146 g $CH_4$ $h^{-1}$ across the US (Townsend-Small et al., 2016) and 35 g $CH_4$ $h^{-1}$ in the UK (Boothroyd et al., 2016). As most of the methods presented here require access to the source, we considered 200 g $CH_4$ $h^{-1}$ to be a sensible limit to the emission rate and is larger than the emissions observed by many previous studies.  Therefore, the scope of this study is limited to estimating $CH_4$ emissions from a single point source that we would realistically be able to approach and measure, i.e. between 40 and 200 g $CH_4$ $h^{-1}$."

At L74:

"We add the caveat that we will only present data from measurement methodologies can be conducted safely, wearing PPE as regulated at the Colorado State University Methane Emissions Technology Evaluation Center (METEC) facility in Fort Collins, CO, USA (steel toe boot, FR overalls, hard hat, safety glasses and 4-gas monitor)."

**Reviewer 2 Specific comment 10:**

L108: can you please then comment on what was done at leak rates higher than 200 g/h? What are the limitations to transfer these conclusions to smaller leak rates? Why should we care about leak rates below 200g/h?

Emissions of 200 g $h^{-1}$ are not insignificant and you know they are happening if you stand next to them. Typically emissions of this size would be measured using downwind techniques. From experience, it becomes difficult to work effectively in areas near NG emissions greater than 200 g $CH_4$ $h^{-1}$. As a rule of thumb at this emission rate, we suggest access to areas closer than 10 m of the source be restricted. As most of the methods require access to the source, we considered this to be a sensible limit to emission and it is more than the upper limit to emissions measurements in the field. The maximum emission from unplugged and abandoned wells was measured at 177 g $CH_4$ $h^{-1}$ in West Virginia (Riddick et al., 2019a), 175 g $CH_4$ $h^{-1}$ in Pennsylvania (Pekney et al., 2018), 146 g $CH_4$ $h^{-1}$ across the US (Townsend-Small et al., 2016) and 35 g $CH_4$ $h^{-1}$ in the UK (Boothroyd et al., 2016).

Changes to the manuscript:

At L59:

"From experience and the response of a 4-gas monitor, working close enough to measure emissions greater than 200 g $CH_4$ $h^{-1}$ for many of these methods (especially the chambers and Hi Flow) can be unsafe, therefore this study is limited to quantifying $CH_4$ emissions between the lowest flow METEC can produce (40 g $CH_4$ $h^{-1}$) and the highest flow we feel comfortable measuring with these methods (200 g $CH_4$ $h^{-1}$). Putting these emission ranges into real-word context, the maximum emission from unplugged and abandoned wells was measured at 177 g $CH_4$ $h^{-1}$ in West Virginia (Riddick et al., 2019a), 175 g $CH_4$ $h^{-1}$ in Pennsylvania (Pekney et al., 2018), 146 g $CH_4$ $h^{-1}$ across the US (Townsend-Small et al., 2016) and 35 g $CH_4$ $h^{-1}$ in the UK (Boothroyd et al., 2016). As most of the methods presented here require access to the source, we considered 200 g $CH_4$ $h^{-1}$ to be a sensible limit to the emission rate and is larger than the emissions observed by many previous studies. Therefore, the scope of this study is limited to estimating $CH_4$ emissions from a single point source that we would realistically be able to approach and measure, i.e. between 40 and 200 g $CH_4$ $h^{-1}$."

At L74:

"We add the caveat that we will only present data from measurement methodologies can be conducted safely, wearing PPE as regulated at the Colorado State University Methane Emissions Technology Evaluation Center (METEC) facility in Fort Collins, CO, USA (steel toe boot, FR overalls, hard hat, safety glasses and 4-gas monitor)."

**Reviewer 2 Specific comment 11:**

L110: it seems that the section 2 repeat the method description from the introduction, or at least lists again each method. The text would be more fluid and logical if all method description (including Eqs 1-5) are into the Method section and the introduction then focuses more strongly on state of the art, context and research questions.

Response to reviewer:

As suggested have moved introduction to the methods section

**Reviewer 2 Specific comment 12:**

L117: what are the uncertainties associated to the release rate (for example from the mass flow controllers)? How long does the releases last? What is the shape of the injection exhaust/outlet? Is there any attempt to reproduce a 'diffusive' exhaust? Or to control the gas exhaust velocity?

Response to reviewer:

At the METEC site, compressed natural gas, with methane compositions ranging from 85 to 95%vol, is supplied from two 145 L cylinders. Flown rates are controlled using a pressure regulator and precision orifices. A point source is considered to be an emission from an aperture. In this case, the hole was a 6mm diameter tube. We have included this detail in the methods section.

Changes to the manuscript:

At L86:

"(diameter 6 mm)"

At L82

"At the METEC site, compressed natural gas, with methane compositions ranging from 85 to 95%vol, is supplied from two 145 L cylinders and flow rates controlled using a pressure regulator and precision orifices."

**Reviewer 2 Specific comment 13:**

L122: how long is the calibration? How precise is the gas standard?

Response to reviewer:  The calibration takes the same time as a sample to run through the GC ~ 20 minutes.  The gas sample accuracy is rated at ± 5%.

Changes to the manuscript:

At L93

"(accuracy of standard ± 5%)"

**Reviewer 2 Specific comment 14:**

L127: how well do we expect the air inside the chamber to be mixed with the fan?

Response to reviewer:

We expect the air to be completely mixed, this is a major source of uncertainty with the statice chamber.

Changes to the manuscript:

L103:

"A fan was secured inside the chamber and used to circulate the air following the method of (Riddick et al. 2019a) to ensure the air inside the chamber was fully mixed."

**Reviewer 2 Specific comment 15:**

L128 how is the air sample drawn?

Response to reviewer:

Using a gas syringe.

Changes to the manuscript:

L106:

"When the chamber is sealed with the ground, following Riddick et al. (2019a), an air sample is drawn using a gas syringe."

**Reviewer 2 Specific comment 16:**

L134: why zero wind condition and not "wind speed below X m/s"? can you elaborate on this serious limitation?

Response to reviewer:

"zero wind conditions" was used to represent the absence of wind (which happens often at METEC). During any wind the chamber acted as a sail, smaller chambers are better in the wind but more dangerous as they fill with gas quickly.

Changes to the manuscript:

At L113:

"During any wind the chamber acted as a sail and larger chamber lifted from the ground, therefore, smaller chambers are better in the wind but quickly fill with gas making measurement difficult."

**Reviewer 2 Specific comment 17:**

L165: at what height is the gas scooter measurement made? Was any attempt made to measure CH4 across the plume (cross wind) to confirm the gaussian shape of the plume? If not, why?

Response to reviewer:

1.5 m. Measuring the shape and size of a plume is non-trivial even in neutral atmospheric stability conditions and difficult to do with plumes as small as presented here as very small changes in wind direction will change the plume. The aim here was to measure the time averaged concentration at a point downwind of the plume.

**Reviewer 2 Specific comment 18:**

L178: What is the impact of very small distance (5m) on the accuracy – does the model have a lower limit? Also measurements at 5m downwind suggest that relatively close access (and at the same height AGL) is possible, this should be acknowledged (also in Table 1).

Response to reviewer:

The model does not technically have any minimum limiting caveats about distance.

Changes to the manuscript:

At L192:

"Here, it assumed that the experiments are conducted as close as possible to the source without direct access to the emission point."

Table 1 caption

"(Y denotes having permission to touch/enclose the emission point and N denotes experiments are conducted as close as possible to the source without direct access)"

**Reviewer 2 Specific comment 19:**

L195 and following: why not automate chamber opening to avoid explosive limit? This should be easily done in a commercial context, and acknowledged in the discussion.

Response to reviewer:

This could be done, but the chamber is still collecting an unknown composition of gas. The automation of a chamber then takes away the operator's control of when this is released. The operator taking an unnecessary risk unless they are wearing self-contained breathing apparatus.

Changes to the manuscript:

At L257:

"The static chamber could be automated to release gas when $CH_4$ concentration inside the chamber approaches LEL to prevent chamber becoming explosive. The major shortcoming of this strategy is that the automation of a chamber takes away the operator's control of when gas is released, which could happen at an inconvenient during measurement. If an automated system is used for collecting gas of unknown composition self-contained breathing apparatus should be worn."

**Reviewer 2 Specific comment 20:**

L227: the accuracy of the single "snapshot" becomes irrelevant once 3 repeat measurements are available for the release. Accuracy and precision derived from 3 points supersede the single measurement discrepancy as informative numbers.

Response to reviewer:

We acknowledge this may be of limited interest and have removed the section and have discussed $A_r$.

Changes to manuscript

At L317:

"Both the dynamic chamber ($A_r$ = -10%, -8%, -10% at emission rates of 40, 100 and 200 g $CH_4$ $h^{-1}$, respectively) and Hi Flow ($A_r$ = -18%, -16%, -18%) repeatedly underestimate the emission, but the dynamic chamber is the most accurate for measurement. For the far field methods, the bLs method underestimated emissions ($A_r$ = +6%, -6%, -7%) while the GP method significantly overestimated the emissions ($A_r$ = +86%, +57%, +29%) despite using the same meteorological and concentration data as input. These findings are consistent with another study (Bonifacio et al., 2013), however, this is the first study that has compared both to a known emission rate. In all cases the accuracy in

the emission estimate increased with emission rate apart from the Hi Flow.  The Bacharach Hi Flow system is designed to measure emission from 50 g $CH_4$ $h^{-1}$ to 9 kg $CH_4$ $h^{-1}$ to an accuracy of ± 10%.  All flow rates presented here are at the lowest range that the Hi Flow can measure and it is likely that the uncertainty in the systems sensors that measures between 40 and 400 g $CH_4$ $h^{-1}$ is of negligible difference.

The method that improves the most as the emission rate increases is the GP method, where accuracy increases from +87% to +29% as the emission rate increased from 40 to 200 g $CH_4$ $h^{-1}$.  This improvement in emission is likely caused by the increased size of the plume and the ability of GP model to parameterize the concentration at distances from the centerline of the plume.  Although not explicitly stated, the parameterization of the lateral dispersion in the GP model is the same at 100 m as at 5 m which is unlikely.  Other controlled release experiments using the GP approach show similar uncertainties, one experiment reported average emissions calculated using a GP model less than 20% (release rates were not published), with the uncertainty mainly driven by atmospheric variability (Caulton et al., 2019).  Another showed uncertainties of ±50% for triplicate measurements of emissions between 90 and 970 g $CH_4$ $h^{-1}$ (Caulton et al., 2018).

Data do not exist on controlled release experiments using a dynamic chamber.  One study suggested a theoretical emissions uncertainty in the dynamic chamber approach of ±7% (Riddick et al., 2019a), with the largest source of uncertainty caused by the measurement of the flow rate of air through the chamber.  Other sources of uncertainty for the dynamic chamber methods are relatively negligible as the methane quantification of the background gas and the gas at steady state (assuming complete mixing of the gas in the chamber) using the GC is highly accurate over a large concentration range and the volume of the chamber fixed by a plastic structure.

A controlled release has been conducted for the bLs model, but only for an emission from an area source (Ro et al., 2011) at the surface and not analogous to the emissions of this study.  Ro et al. (2011) estimated the bLs uncertainty at ± 25% for a gas emitted at an unspecified rate from a 27 $m^2$ emission area.  As with the GP approach, the bLs model's main source uncertainty is the parameterization of the atmospheric stability (Riddick et al., 2012; Flesch et al., 1995; Ro et al., 2011).  The main advantage of the bLs model over the GP at these short distances is it calculates the lateral dispersion of gas for individual particles, while the GP uses an averaged dispersion parameter."

**Reviewer 2 Specific comment 21:**

L232: abs(A) "decreases": this is not confirmed in Fig 2b. can you please explain where this comes from?

Response to reviewer:

Have removed the section

**Reviewer 2 Specific comment 22:**

L242: please quantify "generally"

Response to reviewer:

Have removed the sentence

**Reviewer 2 Specific comment 23:**

L245: Ar>A : to me this is meaningless. It just means that there was some luck in the first value, and therefore it carries little sense to report A once you have Ar.

Response to reviewer:

As suggested, have removed the sentence.

**Reviewer 2 Specific comment 24:**

L251: This is expected and may be seen as trivial. On the opposite, what is surprising is when it is not the case. Why does SD for some techniques increase with increasing emission rate?

Response to reviewer:

Have added text to discuss the S.D.

Changes to the manuscript:

At L357:

"The emission estimates quantified using direct methods, dynamic chamber and Hi Flow sampler, have a lower S.D. than the far-field methods (Figure 2B). The S.D. of direct measurement methods remain relatively constant for emissions between 40 and 200 g $CH_4$ h$^{-1}$ and reflects the relative simplicity of the methods. Assuming all other

parameters are measured correctly, for direct methods the variability in emission estimate is a function of how well the $CH_4$ is mixed into the air in the chamber during the measurement.

Variability in the far field emission estimates is much larger an reflects the relative complexity of inferring emissions. Variability in wind speed, wind direction and atmospheric stability over the 20-minute averaging period are likely to propagate through to large variability in the emission estimate. It may be reasonable to suggest that the variability in bLs calculated emission less than for the GP method because of the added parametrization available (roughness length and gas species). In summary, the penalty of downwind measurement is a higher uncertainty in individual measurements, but this appear to be corrected for by the bLs model through repeat measurements where uncertainty is corrected for by the stochastic nature of particle movement modelling."

**Reviewer 2 Specific comment 25:**

Fig.3 in my opinion would deserve further comments and discussion.

Response to reviewer:

Have moved Figure to become Figure 2B, as suggested, and added text to discuss.

Changes to the manuscript:

At L346:

"The emission estimates quantified using direct methods, dynamic chamber and Hi Flow sampler, have a lower S.D. than the far-field methods (Figure 2B). The S.D. of direct measurement methods remain relatively constant for emissions between 40 and 200 g $CH_4$ h$^{-1}$ and reflects the relative simplicity of the methods. Assuming all other parameters are measured correctly, for direct methods the variability in emission estimate is a function of how well the $CH_4$ is mixed into the air in the chamber during the measurement.

Variability in the far field emission estimates is much larger an reflects the relative complexity of inferring emissions. Variability in wind speed, wind direction and atmospheric stability over the 20-minute averaging period are likely to propagate through to large variability in the emission estimate. It may be reasonable to suggest that the variability in bLs calculated emission less than for the GP method because of the added parametrization available (roughness length and gas species). In summary, the penalty of downwind measurement is a higher uncertainty in individual measurements, but this appear to be corrected for by the bLs model through repeat measurements where uncertainty is corrected for by the stochastic nature of particle movement modelling."

**Reviewer 2 Specific comment 26:**

L265: Which technique has the lowest A is relatively unimportant when Ar is available. Ar>A is also fairly trivial from a very general perspective.

Response to reviewer:

As suggested, have removed the sentence.

**Reviewer 2 Specific comment 27:**

L280-283: These sentences are vague, complicated. There seems to be a confusion between the site scale uncertainty and the single GP uncertainty. Could you please clarify in terms of separating biases and random errors? (and all methods here seem to have consistent biases). Did you make any attempt to look at uncertainty budget in the different methods, the GP in particular, including the choice of stability class? Does it match with the 3-point accuracy?

Response to reviewer:

We have added to this section to investigate the causes of the inaccuracy of the GP model.

Changes to the manuscript:

At L363:

"It is, however, concerning that many of the methods show a bias in measurement results and in particular the GP model (Figure 3). In most studies, it is assumed that in taking multiple measurements the average uncertainty will be reduced to an aggregate, unbiased emission estimate. Taking the GP emission estimates as an example, the individual calculated emissions are all overestimates of the true emission, therefore, suggesting a fundamental shortcoming in the method (Figure 3). These measurements were taking four days apart in similar environmental conditions (all PGSC C) with wind direction being the only difference between measurements, which can be seen from the correlation between the uncertainty and horizontal distance from plume center (Figure 3B). As mentioned above, it is likely that this is due to the lateral dispersion in the GP approach being parametrized incorrectly, i.e. using values that were defined for distances of 100 m. This suggests that using the GP approach for distances less

than 100 m, it is not correct to assume that repeat measurements will remove bias in the calculated average emission."

**Reviewer 2 Specific comment 28:**

L293-294: Can you please explain the "meaning and balance" and provide examples of studies that "present unexpected findings"?

Response to reviewer:

Have removed this sentence

**Reviewer 2 Specific comment 29:**

Fig. 4: other parameters/selectors would be useful but are ignored in this study: is the source buried? What is the source intensity? Did you perform a source detection prior to quantification? Some techniques are missing; as such, the value of this diagram is very poor for decision making, although the idea is good.

Response to reviewer

On reflection we have decided to remove the decision making flow chart.

Editorial comments

**Reviewer 2 Editorial comment 1:**

L21: GP: expand acronym

Changes to the manuscript:

At L 11:

"(GP)"

**Reviewer 2 Editorial comment 2:**

L103: "copy" I suggest "reproduce"

Response to reviewer:

Changed as suggested

**Reviewer 2 Editorial comment 3:**

L177-178: the cited papers are not listed in the References section

Response to reviewer:

Added to Reference Section

**Reviewer 2 Editorial comment 4:**

Fig 3: what is y axis unit?

Response to reviewer:

Added "(%)" to Figure 3

---

## Author Response (AR2)

The Powerhouse Energy Campus
Colorado State University
430 North College Avenue
Fort Collins, CO 80524
Tel: (970) 213-1984

E-mail: Stuart.Riddick@colostate.edu

30th June 2022

Dear Dr Chen,

This letter is to accompany the resubmission of our manuscript entitled "*A quantitative comparison of methods used to measure smaller methane emissions typically observed from superannuated oil and gas infrastructure*", which we would like you to consider for publication. Our paper describes controlled release experiments at the METC facility in Fort Collins, USA that investigate the accuracy and precision of several methods commonly used to measure methane emissions. The controlled releases were all below 200 g $CH_4$ $h^{-1}$ and the methods include: static chambers, dynamic chambers, a Hi-Flow sampling system, a backward Lagrangian stochastic method and the Gaussian Plume method. To our knowledge this is the first time that methods for measuring methane emissions from point sources between 40 and 200 g $CH_4$ $h^{-1}$ have been quantitively assessed against a known reference source and each other.

We appreciate your time in reviewing our manuscript and look forward to hearing from you.

Kind regards,

Stuart N. Riddick (corresponding author)

and co-authors: Riley Ancona, Clay Bell, Mercy Mbua, Aidan Duggan, Tim Vaughan, Kristine Bennett and Dan Zimmerle

Dear Dr Chen,

We thank reviewer 1 for their comments. As suggested, we have amended the manuscript to address the reviewers' comments and have indicated changes to the manuscript in red text.

Please find our detailed responses below.

**Reviewer 1 General comment 1:**

First, it's unclear what the authors mean by "small". The emission rates that the authors are looking at (40-200 g/hour methane) are orders of magnitude larger than soil emission rates (e.g., agricultural soils). I am pointing out the agricultural soils because the authors are attempting to replicate methods for agricultural soils, although it's unclear if they are because the method doesn't seem to be the same – in particular the use of the fan and the lack of pre-installation of collars. It seems to be that the methods presented here are really focused on oil and gas wells. I suggest that the authors be clear on the applicability of the results and revise the title accordingly.

Response to reviewer:

As suggested, we have changed the title and text to highlight this study is focused on emissions from oil and gas infrastructure.

Changes to the manuscript:

Title: A quantitative comparison of methods used to measure smaller methane emissions typically observed from superannuated oil and gas infrastructure

At L7: Recent interest measuring methane ($CH_4$) emissions from abandoned oil and gas infrastructure has resulted in several methods being continually used to quantify point source emissions less than 200g $CH_4$ hour$^{-1}$.

At L41: Several methods are being used to measure emissions from these smaller point sources (less than 200 g $CH_4$ hour$^{-1}$) from abandoned oil and gas infrastructure.

**Reviewer 1 General comment 2:**

The authors claim in a few places that the static chamber is the simplest. But based on what's presented, the HiFlow sampler is the simplest – especially in terms of use and post-measurement analysis.

Response to reviewer:

This is a subjective observation that we are happy to remove.

**Reviewer 1 General comment 3:**

It would be easier for the reader if the Conclusions section is separated into two sections: discussions and conclusions.

Response to reviewer:

Have edited the manuscript as suggested

LINE-BY-LINE COMMENTS

**Reviewer 1 Line-by-line comment 1:**

Title: What is meant by "small"? Are the authors referring to the physical size of the emission source or the emission rate? Also, specify that this study applies mainly to oil and gas sources.

Response to reviewer:

As above, have changed the title and some of the text

Changes to the manuscript:

Title: A quantitative comparison of methods used to measure smaller methane emissions typically observed from superannuated oil and gas infrastructure

At L7: Recent interest measuring methane ($CH_4$) emissions from abandoned oil and gas infrastructure has resulted in several methods being continually used to quantify point source emissions less than 200g $CH_4$ hour$^{-1}$.

At L41: Several methods are being used to measure emissions from these smaller point sources (less than 200 g $CH_4$ hour$^{-1}$) from abandoned oil and gas infrastructure.

**Reviewer 1 Line-by-line comment 2:**

line 36: There are missing references here: Townsend-Small et al, 2021, Saint-Vincent et al. 2020, and El Hachem and Kang, 2022.

Response to reviewer:

Have added references as suggested.

**Reviewer 1 Line-by-line comment 3:**

line 38: replace "form" with "inform". Here and elsewhere, the paper needs proofreading.

Response to reviewer:

We believe "form" is correct and have proofread the rest of the manuscript.

**Reviewer 1 Line-by-line comment 4:**

line 48: grammatical error.

Response to reviewer:

Have deleted "using"

**Reviewer 1 Line-by-line comment 5:**

line 51-52: Tracer release has been used to measure active and abandoned wells in Romania. See Delre et al. (2022) published in Elementa.

Response to reviewer:

Have added the citation.

Changes to the manuscript:

At L55: Tracer release is technically demanding, takes a long time to make a single measurement and requires road access for measurement, although it has been used to measure nonproducing wells in Hungary (Delre et al., 2022)

**Reviewer 1 Line-by-line comment 6:**

line 72: spell out "PPR"

Response to reviewer:

Have added.

Changes to the manuscript:

At L76 personal protective equipment (PPE)

**Reviewer 1 Line-by-line comment 7:**

line 73: spell out "FR".

Response to reviewer:

Have added to text

Changes to the manuscript:

At L77: flame resistant (FR)

**Reviewer 1 Line-by-line comment 8:**

line 80: the controlled release testing was performed with 85 to 95% by volume methane but if there is uncertainty in the methane concentration, this would translate to an error in the target emission rate. Which concentration was used to determine the target flow rate?

Response to reviewer:

At METEC the methane content of the natural gas in each release is measured by gas chromatography and accounted for in the known emission rate.

Changes to the manuscript:

At L85: At METEC the methane content of the natural gas in each release is measured by gas chromatography and accounted for in the known emission rate.

**Reviewer 1 Line-by-line comment 9:**

line 93: remove "relatively simple" as it is too subjective. The simplest method considered here is the HiFlow sampler.

Response to reviewer:

Have removed

Changes to the manuscript:

At L98: For the static chamber method a container of a known volume

**Reviewer 1 Line-by-line comment 10:**

line 95: it is not necessary to have a fan in a static chamber. Collier et al (2014) did not use a fan and Pihalatie et al. (2013) did not always use a fan.

Response to reviewer:

Acknowledged, have removed the sentence.

**Reviewer 1 Line-by-line comment 11:**

line 99: as mentioned above, Pihalatie et al. (2013) and Collier et al (2014) do not always use a fan. More importantly, their focus is on measuring agricultural soils with emission rates orders of magnitude below the range considered in this paper. The authors need to justify why the methods outlined in these papers are followed given that the application and emission rates are so different.

Response to reviewer:

Have changed the reference to Kang et al (2014; 2016) that did use the static chamber to measure emissions from oil and gas wells.

**Reviewer 1 Line-by-line comment 12:**

line 108: replace "chambers" with "chamber"

Response to reviewer:

Have replaced

**Reviewer 1 Line-by-line comment 13:**

line 110: was the chamber anchored to the ground as done in Collier et al (2014)? Following Collier et al. (2014), the anchoring needs to be done 1 day in advance. Also, why would the static chamber be more likely to be lifted off the ground compared to the dynamic chamber? Is the pump somehow anchoring the dynamic chamber?

Response to reviewer:

The chamber was made of two parts, a smaller lower part that was secured 4 cm into the soil and a larger upper part that was fixed to the lower part at the start of the experiment.

The $0.5 \text{ m}^3$ chamber was very large and easily caught the wind. The dynamic chamber was smaller and had the pump sitting on top.

Changes to the manuscript:

At L104: The chamber was constructed of two parts, a smaller lower part that was secured 4 cm into the soil and a larger upper part that was fixed to the lower part at the start of the experiment.

At L 115: larger $(0.5 \text{ m}^3)$ chamber

At L132: container $(0.12 \text{ m}^3)$ enclosing

**Reviewer 1 Line-by-line comment 14:**

line 141: did Riddick et al (2019) measure at similar rates as those tested in this paper?

Response to reviewer:

Yes, the emissions from abandoned wells reported by Riddick et al. (2019) ranged from background to 100 g $CH_4$ $h^{-1}$

Changes to the manuscript:

At L142: Methane emissions from abandoned wells have been quantified using this method between 4 µg $CH_4$ $hr^{-1}$ and 100 g $CH_4$ $hr^{-1}$ (Riddick et al., 2019a).

**Reviewer 1 Line-by-line comment 15:**

line 143: what is the criteria to determine if steady state has been reached?

Response to reviewer:

As stated at L148, the steady state was reached when the $CH_4$ concentration inside had become constant, as measured by the HXG-2D sensor.

Changes to the manuscript

At L148: The chamber was left until the CH4 concentration inside had become constant, as measured by a Sensit HXG-2D sensor (Sensit Technologies, Valparaiso, IN, USA).

**Reviewer 1 Line-by-line comment 16:**

line 152: is the industry standard set by some industry body? Or do the authors mean that this is common practice?

Response to reviewer:

Have deleted "industry standard"

**Reviewer 1 Line-by-line comment 17:**

line 169-170: how is the PGSC determined for a measurement determined? This is provided to some extent in the supporting information. However, given the large error in the GP method and the authors' attribution of errors to the PGSC parameterization, it is important to discuss them here.

Response to reviewer:

The PGSC can either be calculated using the wind speed and a measure of the solar irradiance (Supplementary Material Section 1 Table S1) or using a sonic anemometer. Here, the former method was employed and the PGSC calculated from the wind speed ($u$, m s$^{-1}$) measured at 1.2 m and irradiance measured at the emission point ($G$, kW m$^{-2}$). Pasquill and Smith (1983) originally defined strong irradiance as sunny midday in midsummer in England and slight insolation to similar conditions in midwinter. Here we class strong irradiance as > 1 kW m$^{-2}$, Moderate irradiance 0.5 kW m$^{-2}$ to 1 kW m-2 and Light irradiance as > 0.5 kW m$^{-2}$.

Changes to the manuscript:

At L191: The PGSC can either be calculated using the wind speed and a measure of the solar irradiance (Supplementary Material Section 1 Table S1) or using a sonic anemometer. Here, the former method was employed and the PGSC calculated from the wind speed ($u$, m s$^{-1}$) measured at 1.2 m and irradiance measured at the emission point ($G$, kW m$^{-2}$). Pasquill and Smith (1983) originally defined strong irradiance as sunny midday in midsummer in England and slight insolation to similar conditions in midwinter. Here we class strong irradiance as > 1 kW m$^{-2}$, Moderate irradiance 0.5 kW m$^{-2}$ to 1 kW m$^{-2}$ and Light irradiance as > 0.5 kW m$^{-2}$.

**Reviewer 1 Line-by-line comment 18:**

line 179: subscript 4 in "CH4".

Response to reviewer:

Have changed

**Reviewer 1 Line-by-line comment 19:**

line 189: provide approximate range of distances rather than saying "as close as possible".

Response to reviewer:

Have included a range of distances

Changes to the manuscript:

At L199: (between 1 and 10 m)

**Reviewer 1 Line-by-line comment 20:**

line 229: The Hi Flow sampler is logistically the simplest, not the static chamber.

Response to reviewer:

As above, have removed the sentence

**Reviewer 1 Line-by-line comment 21:**

line 250: are the authors suggesting that sources emitting aromatic hydrocarbons and hydrogen sulphide should not be used by the static chamber method or by all methods? This sentence is unclear.

Response to reviewer:

Have removed the sentence.

**Reviewer 1 Line-by-line comment 22:**

line 251: It's unclear what "our measurement data" is.

Response to reviewer:

Have changed the sentence

Changes to the manuscript:

At L 258: As such, we have not presented the measurement data collected during the static chamber experiments and strongly encourage the use of an alternative method.

**Reviewer 1 Line-by-line comment 23:**

line 254: missing word. proof read.

Response to reviewer:

Have added "time" to the manuscript.

**Reviewer 1 Line-by-line comment 24:**

line 276: how much does the Hi Flow instrument cost?

Response to reviewer:

The Hi Flow costs $35k.

Changes to the manuscript:

At L284: (costs $35,000)

**Reviewer 1 Line-by-line comment 25:**

line 282: how is safety assessed here? the oil and gas industry has specific safety standards and ways to assess/handle safety, and personnel are required to go through rigorous safety training. All oil and gas sites can be dangerous and carry safety risks. I suggest that the authors not make conclusions on safety conditions on sites, as that's beyond the scope of this work.

Response to reviewer:

Have removed the comment.

Dear Dr Chen,

We thank reviewer 2 for their comments. As suggested, we have amended the manuscript to address the reviewers' comments and have indicated changes to the manuscript in red text.

Please find our detailed responses below.

**Reviewer 2 General comment 1:**

The result section is fairly light and short in substance for a research paper, or at least by the standard that I would expect in AMT.

Response to reviewer:

All of the results are presented in the SI to help reduce the word limit and reduce the burden on the publication. The aim of this study was to produce simple ground-truth validation of quantification techniques and present the data is as simple a form as possible.

**Reviewer 2 General comment 2:**

The Conclusion section is structured in a very unusual way, with a new figure (Fig 3) and comparison against other research that were not introduced before. I suggest to split this Conclusions into a Discussion section and a more "classic" Conclusion section.

Response to reviewer:

Have amended the manuscript as suggested.

**Reviewer 2 General comment 3:**

L50 I suggest to moderate this statement in relation to the leak rate e.g. "cannot be used to quantify emission below XXX kg/h")

Response to reviewer: The sentence is correct as it stands, OGI cameras are not quantification instruments and con only be use for emission detections.

Changes to manuscript:

At L 50: While OGI cameras can be used for detecting emissions greater than 20 g $CH_4$ $h^{-1}$ (Ravikumar et al., 2018; Stovern et al., 2020; Zimmerle et al., 2020), using this method for quantification remains in development with few studies published to date investigating the accuracy of emission rate estimates from OGI (Kang et al., 2022).

**Reviewer 2 General comment 4:**

L361 The use of the GP approach here is made with a single measurement in plume. If this context were explicitly stated I would agree with the statement. However mobile in situ measurements in (and across) the plume, even at distances shorter than 100m, would give much better results.

Response to reviewer:

Have clarified the text with your suggestion.

Changes to manuscript:

At L 369: This suggests that using the GP approach with a single measurement in the plume for distances less than 100 m, it is not correct to assume that repeat measurements will remove bias in the calculated average emission. It is currently unclear if mobile, in-situ measurements in and across the plume, even at distances shorter than 100m, would give much better results.

Editorial comments

**Reviewer 2 Editorial comment 1:**

L25 add against in: and AGAINST each other

Response to reviewer: As suggested, have added "against"

**Reviewer 2 Editorial comment 2:**

L50 difficulty

Response to reviewer: Have corrected

**Reviewer 2 Editorial comment 3:**

L75 add against in: and AGAINST each other

Response to reviewer: As suggested, have added "against"

---

## Author Response (AR3)

The Powerhouse Energy Campus
Colorado State University
430 North College Avenue
Fort Collins, CO 80524
Tel: (970) 213-1984

E-mail: Stuart.Riddick@colostate.edu

14$^{th}$ October 2022

Dear Dr Chen,

This letter is to accompany the resubmission of our manuscript entitled "A quantitative comparison of methods used to measure smaller methane emissions typically observed from superannuated oil and gas infrastructure", which we would like you to consider for publication. Our paper describes controlled release experiments at the METC facility in Fort Collins, USA that investigate the accuracy and precision of several methods commonly used to measure methane emissions. The controlled releases were all below 200 g $CH_4$ $h^{-1}$ and the methods include: static chambers, dynamic chambers, a Hi-Flow sampling system, a backward Lagrangian stochastic method and the Gaussian Plume method. To our knowledge this is the first time that methods for measuring methane emissions from point sources between 40 and 200 g $CH_4$ $h^{-1}$ have been quantitively assessed against a known reference source and each other.

We appreciate your time in reviewing our manuscript and look forward to hearing from you.

Kind regards,

Stuart N. Riddick (corresponding author)

and co-authors: Riley Ancona, Clay Bell, Mercy Mbua, Aidan Duggan, Tim Vaughan, Kristine Bennett and Dan Zimmerle

We thank the reviewer for their comments. As suggested, we have amended the manuscript to address the reviewers' comments and have indicated changes to the manuscript in red text.

Please find our detailed responses below.

**Reviewer 1 General comment 1:**
The paper presents controlled release experiments of multiple methane emission measurement methods that have been used to quantify emission rates of oil and gas sources. They considered the dynamic chamber, the hi flow sampler, the Gaussian plume method, and the backward Lagrangian stochastic models and to some extent, static chambers (see below for more on this). The contribution can be useful as methane monitoring is important for greenhouse gas emission reductions. However, below are some important revisions that are needed to make it easier for readers to understand the paper and the results.

Response to reviewer:
The controlled release experiments conducted at METEC show the static method to be inherently dangerous as we were unable to remove the chamber without the four-gas monitor, worn on the observer's collar, detecting $CH_4$ concentrations that exceeded the lower explosive limit, i.e. triggered the monitor's alarm. This poses a considerable risk to the observer. In addition to being an explosive risk, natural gas emitted from the subsurface could contain aromatic hydrocarbons and other toxic gases which could also collect to hazardous concentrations inside the static chamber and inhaled when removing the chamber. Therefore, we recommend that the static chamber method should not be used to quantify emissions from oil and gas infrastructure. While the reviewer feels that we should omit the static chamber method from the paper completely, we feel that its omission could appear to be a tacit acknowledgement that it is an acceptable method for quantifying emissions. Our observations show the static chamber method to be dangerous and want to clearly state that we do not recommend its use in the field. To make this point we have added the following to the manuscript.

Changes to the manuscript:
L248: "The method is inherently dangerous as we were unable to remove the chamber without the four-gas monitor, worn on the observer's collar, detecting $CH_4$ concentrations that exceeded the lower explosive limit, i.e. triggered the monitor's alarm."
L309: "Static chamber results are not presented as we were unable to remove the chamber without exposing the observer to an explosive environment."
At L 327: "This study investigates the utility, accuracy and precision of five methods"
At L329: "When the method has been shown to be no danger to the observer, we generate $CH_4$ emission estimates"
At L 333: "The static chamber method was found to be inherently dangerous, as the observer was unable to remove the chamber without being exposed to an explosive environment. As a result, the data from the static chamber experiments have not been presented in this study. Furthermore, the experiment conducted at METEC used processed natural gas where heavier/aromatic hydrocarbons and toxic gases have been removed. Gas emitted from abandoned oil and gas wells is unrefined and we advise that the static chamber method should not be used to quantify emissions of an unknown composition of natural gas as this

could expose the observer to high concentrations of toxic gas. Therefore, we recommend that one of the other methods presented here should be used to quantify emissions from abandoned oil and gas wells."

**Reviewer 1 General comment 2:**
There are several structural issues with the paper. Section 3.1 on the Method narrative, which is the first section of the results, would be best placed in the Methods or Discussion sections as no results are presented.

Response to reviewer:
Even though Section 3.1 contains no quantitative data, it presents qualitative information of the suitability of each method for deployment in the field. This helps to address objective 4: "Make recommendations on the suitability of each method for measuring emissions from relatively small point sources.". We feel that it is best placed in the results section. To highlight this, we have changed the name of the section to: "3.1 Method narrative – Qualitative observations of methods"

Changes to the manuscript:
At L 242: "**3.1 Method narrative – Qualitative observations of methods"**

**Reviewer 1 General comment 3:**
The presentation of the static chamber in the methods section (and the method narrative) is confusing as no results are shown. It would be easier for readers if the authors just limited the scope to the four methods that they analyzed. The authors could simply say that the four chosen methods fit the authors' objectives and move on.

Response to reviewer:
As noted above (General comment 1), we feel that the inclusion of the static method to this paper is essential. The method is inherently dangerous, should not be used to collect an unknown composition of natural gas at an unknown rate and another method should be used. We note that reviewer is confused by the omission of data and to address this we make our message very clear at the start of each results sections (3.1 and 3.2) and in the discussion.

Changes to the manuscript:
L248: "The method is inherently dangerous as we were unable to remove the chamber without the four-gas monitor, worn on the observer's collar, detecting $CH_4$ concentrations that exceeded the lower explosive limit, i.e. triggered the monitor's alarm."
L309: "Static chamber results are not presented as we were unable to remove the chamber without exposing the observer to an explosive environment."
At L 327: "This study investigates the utility, accuracy and precision of five methods"
At L329: "When the method has been shown to be no danger to the observer, we generate $CH_4$ emission estimates"
At L 333: "The static chamber method was found to be inherently dangerous, as the observer was unable to remove the chamber without being exposed to an explosive environment. As a result, the data from the static chamber experiments have not been presented in this study. Furthermore, the experiment conducted at METEC used processed natural gas where heavier/aromatic hydrocarbons and toxic gases have been removed. Gas emitted from

abandoned oil and gas wells is unrefined and we advise that the static chamber method should not be used to quantify emissions of an unknown composition of natural gas as this could expose the observer to high concentrations of toxic gas. Therefore, we recommend that one of the other methods presented here should be used to quantify emissions from abandoned oil and gas wells."

**Reviewer 1 Line-by-line comment 1:**
L27-28: The GWP for methane is wrong. It's ~30 for a 100 year time frame and is 86 for a 20 year time frame. Also, should cite the latest IPCC assessment report.

Response to reviewer:
As suggested, number changed and citation added

**Reviewer 1 Line-by-line comment 2:**
L95: add "methane" after "5,000 ppm"

Response to reviewer:
"$CH_4$" added

**Reviewer 1 Line-by-line comment 3:**
L109 and L118: One says at least three and then it says four samples. Sounds like redundant statements.

Response to reviewer:
Have deleted "at least three further"

**Reviewer 1 Line-by-line comment 4:**
L135: Would be useful to list the period of time needed for the CH4 concentrations to be stable.

Response to reviewer:
This varies as the time is a function of the emission rate and the flow of air through the chamber and the size of the chamber, so impossible to include a value.

**Reviewer 1 Line-by-line comment 5:**
L135: how did the authors determine that steady state has been reached?

Response to reviewer:
As stated at L150: "The chamber was left until the $CH_4$ concentration inside had become constant, as measured by a Sensit HXG-2D sensor (Sensit Technologies, Valparaiso, IN, USA)."

**Reviewer 1 Line-by-line comment 6:**
L139: which area of the chamber? Footprint?

Response to reviewer:
Changed "area" to "footprint"

**Reviewer 1 Line-by-line comment 7:**
L153: Add "rate" after "emission"

Response to reviewer:
"rate" added

**Reviewer 1 Line-by-line comment 8:**
L157: Is the Hi Flow sampler offered by Heath the same model as the Bacharach Hi Flow Sampler tested here? The names are different. On the Heath website, it doesn't mention the word "Bacharach".

Response to reviewer:
Apologies, this should be "(Bacharach, Pittsburgh, USA, www.mybacharach.com)". Bacharach are the only current manufacturer, Heath does not make a Hi Flow.

**Reviewer 1 Line-by-line comment 9:**
L158: "currently-available"

Response to reviewer:
Changed as suggested

**Reviewer 1 Line-by-line comment 10:**
L159: Connolly et al. (2019) used the Bacharach Hi Flow Sampler and it is not clear that this is the same as the one available from Heath.
Equation 3: all variables in the equation need to be defined.
2.3 Hi Flow. There are three high flow samplers mentioned here: the Bacharach, the Heath Hi Flow Sampler, and the new Hi Flow Unit by Vaughn et al. Which one was used here?

Response to reviewer:
The Bacharach Hi Flow was used in this study. We have added text to clarify this.

Changes to the manuscript:
At L160: "The Bacharach Hi Flow Sampler (Bacharach, Pittsburgh, USA, www.mybacharach.com) is the only currently-available Hi Flow sampler and was used in this study,"
At L169: "The Bacharach Hi Flow sampler used in this study was calibrated monthly as recommended by the manufacturer."

**Reviewer 1 Line-by-line comment 11:**
L165-167: the last sentence is redundant as it's a repeat of what is said two sentences ago.

Response to reviewer:
Deleted as suggested

**Reviewer 1 Line-by-line comment 12:**
L193: how much better would the results be if a sonic anemometer was used? Why wasn't it used?

Response to reviewer:
PGSCs are very granular and the classifications are cover a broad range of Monin-Obukhov lengths, as measured by the sonic anemometer, therefore this is not expected to be a large source of error. Due to power requirements, sonic anemometers are unlikely to be used in the field and, as such, a more basic approach is adopted.

Changes to the manuscript:
At L197: "Due to power requirements, sonic anemometers are unlikely to be used in the field and, as such, a more basic approach is adopted and"

**Reviewer 1 Line-by-line comment 13:**
L196: First "class" should be replaced with "classify". More importantly, what is the justification for using this classification. There is no reference here

Response to reviewer:
"classify" has been added, as has a citation.

**Reviewer 1 Line-by-line comment 14:**
L231: Does twice mean that it was conducted 6 times in total (3 x 2)?

Response to reviewer:
No, the sentence is a legacy of previous edits and is no longer required. It has been removed.

**Reviewer 1 Line-by-line comment 15:**
L238 onwards: The method narrative is not really a result. It should go in the Methods or Discussion. I think a lot of it could just be deleted.

Response to reviewer:
Even though Section 3.1 contains no quantitative data, it presents qualitative information of the suitability of each method for deployment in the field. This helps to address objective 4: "Make recommendations on the suitability of each method for measuring emissions from relatively small point sources.". We feel that it is best placed in the results section. To highlight this, we have changed the name of the section to: "3.1 Method narrative – Qualitative observations of methods"

**Reviewer 1 Line-by-line comment 16:**
L239: this paragraph and the following ones on the static chamber should go in the Methods. Better yet, as per my comment above, these paragraphs should just be deleted.

Response to reviewer:
As mentioned above, we find the result that the method is dangerous warrants its inclusion into the paper. This has been emphasized throughout.

Changes to the manuscript:

L248: "The method is inherently dangerous as we were unable to remove the chamber without the four-gas monitor, worn on the observer's collar, detecting $CH_4$ concentrations that exceeded the lower explosive limit, i.e. triggered the monitor's alarm."

L309: "Static chamber results are not presented as we were unable to remove the chamber without exposing the observer to an explosive environment."

At L 327: "This study investigates the utility, accuracy and precision of five methods"

At L329: "When the method has been shown to be no danger to the observer, we generate $CH_4$ emission estimates"

At L 333: "The static chamber method was found to be inherently dangerous, as the observer was unable to remove the chamber without being exposed to an explosive environment. As a result, the data from the static chamber experiments have not been presented in this study. Furthermore, the experiment conducted at METEC used processed natural gas where heavier/aromatic hydrocarbons and toxic gases have been removed. Gas emitted from abandoned oil and gas wells is unrefined and we advise that the static chamber method should not be used to quantify emissions of an unknown composition of natural gas as this could expose the observer to high concentrations of toxic gas. Therefore, we recommend that one of the other methods presented here should be used to quantify emissions from abandoned oil and gas wells."

**Reviewer 1 Line-by-line comment 17:**
L246-248: the static chambers (referred to as non-steady-state chambers) can be vented or not vented as described in Livingston and Hutchinson, a source the authors here reference.

Response to reviewer:
The action described here is that the analyser outlet is left to the open air, therefore actively pumping air from the chamber (i.e. a dynamic chamber). The word "vented" maybe misleading and has been changed

Changes to the manuscript:
L252: "air from the analyzer exhaust is actively pushed outside the chamber"

**Reviewer 1 Line-by-line comment 18:**
L249: why would the gas coming out of an analyzer be necessarily lower concentration? The analyzer can be non-destructive.

Response to reviewer:
Gas is taken into the analyser, the concentration analysed and a finite time later (depending on the length of the tube and the specification of the instrument) the gas is passed back into the static chamber. We are not suggesting the analyser is destroying the $CH_4$, more the gas reintroduced to the chamber is of a lower concentration as it has been removed and recycled.

**Reviewer 1 Line-by-line comment 19:**
L252-253: Delete sentence beginning with "It is unlikely that gas will mix... ". This statement does not below in the results as there is no result that the authors are presenting to support this.

Response to reviewer:

Have deleted as suggested.

**Reviewer 1 Line-by-line comment 20:**
L279: What gas/air is being pumped? What is the composition of the gas/air and how could it affect results?

Response to reviewer:
The air is ambient drawn through 2 m of tubing and unlikely to affect the results as the source of emission is inside the chamber and far away from the gas vented from the chamber.

Changes to the manuscript:
At L289: "Another factor that could affect accuracy of measurement is the air being pumped into the chamber, care should be taken to ensure the inlet is apart from other $CH_4$ sources and far away from the chamber outlet."

**Reviewer 1 Line-by-line comment 21:**
L318: This sentence contradicts L295, which says bLs is the best.

Response to reviewer:
Acknowledged, this is misleading and have corrected the text.

Changes to the manuscript:
At L341: "but the dynamic chamber is more accurate".